# $S^2$-FracMix: Self-Saliency Fractal Mixup

## ABSTRACT

Data augmentation methods have shown impressive performance in learning train-ing data distributions to minimize the generalization gap. Recently, these ap-proaches have been replaced by adversarial mixup methods to produce online mixed samples to improve robustness and generalization of deep neural networks. In addition, previous saliency-based methods simply extract the salient region from the source image and paste it into target image. Although these approaches improve performance, they may introduce unreliable samples during training in addition to substantial computational overhead. In this paper, we introduce a Self-Saliency ($S^2$) mixup method that creates challenging samples by extracting only salient patches at varying scales and places back into the non-salient regions of the same image. The aim is to learn scale-invariant features to improve generalization with less computational overhead. Also, to improve resilience against adversarial per-turbations, we propose a new approach *FracMix* which only mixes self-similarity pattern into salient patches with different mixing ratios. Our proposed $S^2$-FracMix enables the model to learn from both fractal and non-fractal structures simultane-ously within a single training image, offering a more targeted and label-consistent form of augmentation. The proposed $S^2$-FracMix demonstrates state-of-the-art performance on seven datasets including coarse and fine-grained classification, robustness, calibration, contrastive learning, object detection, few-shot (5, 10, and 100 shots), and transfer learning compared to the existing state-of-the-art methods.

## 1 INTRODUCTION

The exponentially growing size of Deep Neural Networks (DNNs) and excessive representation capabilities have enabled neural networks to fit to a given training data Zhang et al. (2018); Cao et al. (2024); Carratino et al. (2022). To further increase the generalization performance, data augmentation has become an important research direction in machine learning (ML) Kang & Kim (2023); Kim et al. (2020a; 2021). It has been applied to a wide variety of underlying tasks, including image classification Qin et al. (2025); Chen et al. (2022), object detection Zoph et al. (2020), and segmentation Ghiasi et al. (2021); Jin et al. (2025). Due to these characteristics, data augmentation methods reduce the generalization gap on unseen data, prevent model collapse Kang & Kim (2023); Xiao et al. (2023); Wang et al. (2024) and handle distribution shifts Pinto et al. (2022); Jin et al. (2024).

An important aspect of these methods is the improvement in the diversity and robustness of neural networks while maintaining the structural integrity of the data Huang et al. (2023); Han et al. (2022b); Hendrycks et al. (2020); Verma et al. (2019). Another key parameter is the practical utilization while balancing performance and computational overhead Kim et al. (2021; 2020a). In this work, we propose $S^2$-FracMix which improves generalization by encouraging more diversity and structural complexity in the augmented samples while incurring low computational overhead (see Figure 2).

Task-independent *mixup* methods linearly interpolate random pairs of data Zhang et al. (2018). In this domain, other methods such as CutMix Yun et al. (2019), Manifold Mixup Verma et al. (2019), AlignMixup Venkataramanan et al. (2022) and ResizeMix Qin et al. (2020) (see Figure 1) have also been introduced to mix random pairs of data by various ways. These methods create previously unseen virtual examples to improve the generalization and robustness of neural networks. These approaches mix random pair of data and do not preserve salient regions. To overcome this problem, saliency-based methods are proposed to preserve salient regions including SaliencyMix Uddin et al. (2020), PuzzleMix Kim et al. (2020a), Co-Mixup Kim et al. (2021), and GuidedMixup Kang & Kim (2023). In these methods, high computational overhead is unavoidable, adding additional training time

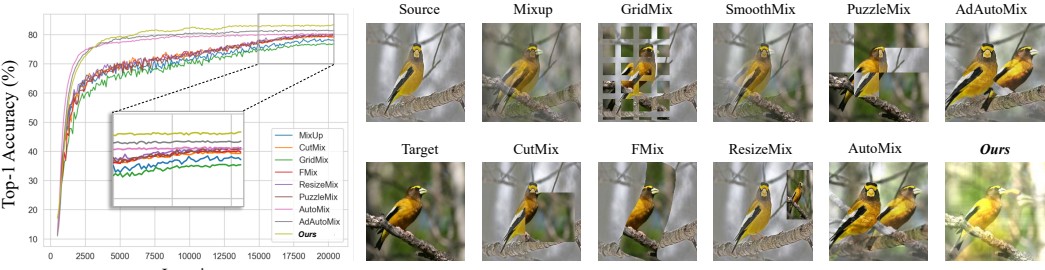

Figure 1: (a) Performance of ResNet-18 trained on $S^2$-FracMix with 200 epochs on CIFAR100. (b) Illustration of $S^2$-FracMix: salient regions are extracted at multiple scales, transformed, enriched with the proposed *FracMix*, and blended back into the original image.

and large-scale computational resources Kim et al. (2020a; 2021). Instead of using fixed heuristics like traditional Mixup Guo et al. (2019), AutoMix Liu et al. (2022b), AdAutoMix Qin et al. (2024) automatically learn how to mix samples and labels.

Despite the introduction of a large number of data augmentation methods, it remains an open question to design an adversarial mixup method that is both memory-efficient and supports multiple mixing modes. Although some researchers proposed adversarial mixup methods Qin et al. (2024); Liu et al. (2022b), often incur substantial training overhead and have not been particularly designed for Vision Transformer architectures. Another key issue of fractal-based adversarial methods Islam et al. (2024a); Huang et al. (2023); Hendrycks et al. (2022) is their blending of self-similar fractals across the entire image, which disturbs the required clean content and induces a distribution shift away from the original data. Consequently, there is a need for a more targeted and efficient approach that exploits self-similar fractals without sacrificing generalization performance.

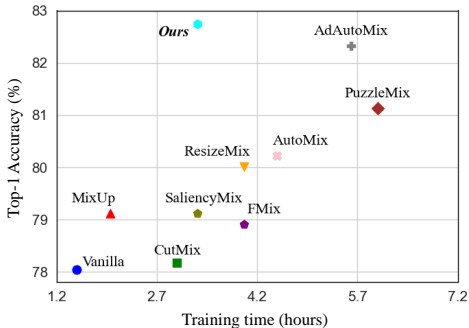

Figure 2: Comparisons of total training time *vs.* top-1 accuracy of `ResNet-18` on CIFAR-100 dataset with RTX 3090 (24GB VRAM). Implementation details are provided in Appendix B.

To this end, we propose $S^2$-FracMix method comprising of two components including Self-Saliency ($S^2$) and *FracMix*. In ($S^2$), we extract multi-scale, saliency-guided patches from an input image, apply different transformations to these patches, and blend them back into the image via a controlled mixing operation. In contrast to DiffuseMix Islam et al. (2024a), which blends fractal textures across the entire image, *FracMix* restricts self-similar fractal injection to the salient patches identified by $S^2$ and incorporates them into the original image. In this way, salient patches with fractal blending increases diversity while preserving semantics, because each training sample simultaneously contains fractal and non-fractal structure. Prior work suggests that fractal injections aid safety and resilience against adversarial perturbations Huang et al. (2023); Hendrycks et al. (2022) by reducing overfitting and improving more diversity and generalization. In addition to these, we also incorporate multiple modes of mixing rather than a fixed recipe which helps model to robustly recognize the object and generalize well to the test set. Extensive experiments show that $S^2$-FracMix outperforms state-of-the-art (SOTA) augmentation baselines in clean accuracy, adversarial robustness, recognition under occlusion and in more scenarios. The main contributions of this work are as follows.

- We propose Self-Saliency ($S^2$) based mixing, that is multi-scale, saliency-guided patches are extracted from an input image, different transformations are applied to these patches, and blended back into the same image via a controlled mixing mechanism.

- We propose *FracMix*, which mix self-similar fractal structure only within saliency-guided patches, preserving clean context while increasing structural complexity. This targeted mixing yields

images that contain both fractal and non-fractal patterns simultaneously, improving robustness and ML safety without sacrificing data fidelity.

- We introduce a high-level mixing of multiple mixing modes by random selection for each training instance to make the model more generalizable.

- Experimental results on seven datasets and comparisons with nine SOTA methods demonstrate that our models trained with $S^2$-FracMix consistently improve performance on a variety of tasks, including general and fine-grained classification, object detection, transfer learning, self-supervised learning, calibration, and few-shot learning (10, 50, 100 shots).

## 2 RELATED WORK

Some earlier work in the field of data augmentation is mentioned in Appendix Section C.1 and generative based data augmentation methods are presented in Appendix Section C.2.

**Mixup Augmentation** Mixup methods obtain more reliable augmented samples Lee et al. (2020); Yang et al. (2022); Hong et al. (2021). Manifold Mixup (Verma et al., 2019) extends this interpolation in hidden layers, thereby improving latent representations. CutMix (Yun et al., 2019) replaces rectangular patches between two images to promote occlusion-aware robust learning. A sequence of methods such as FMix Harris et al. (2020), GridMix Baek et al. (2021) use similar methods, whereas ResizeMix Qin et al. (2020) resizes a patch from one image and overlays it onto another to achieve scale-aware transformations. SnapMix (Huang et al., 2021) proportionally mixes semantically relevant patches for fine-grained classification. Decoupled Mixup Liu et al. (2024) proposed an efficient mixup objective function with a decoupled regularizer by using hard mixed samples to mine discriminative features.

**Automated Mixup Augmentation** These methods focus on a trade-off between mixing strategies and optimization complexity, since image mixing is disconnected from the training task. To overcome this, AutoMix Liu et al. (2022b) proposes a framework that jointly optimizes mixed sample generation and classification, ensuring continuous creation of relevant samples. Recent advancements include adversarial data augmentation Zhao et al. (2020) and GAN-based methods Antoniou et al. (2017) aim to automate augmentation. Adversarial MixUp Qin et al. (2024) addresses domain shift by synthesizing mixed samples for adaptation.

**Advarsarial Mixup Methods** Some influential line of work integrates fractal images directly into the augmentation pipeline to improve model safety. PixMix Hendrycks et al. (2022) augments training images by blending them with synthetic images including fractals and feature visualizations. Building on this, IPMix Huang et al. (2023) introduces multi-scale fractal mixing, where fractal patterns are inserted at pixel, patch, and image levels. Recent studies have further combined fractal augmentation with generative models. DiffuseMix Islam et al. (2024a) blends a diffusion generated image with the original, then overlays a fractal image, resulting in augmented views that are structurally complex.

**Saliency-driven Mixup Augmentation** Saliency-based methods such as SaliencyMix Uddin et al. (2020) and Attentive-CutMix Walawalkar et al. (2020) mixed the most discriminative regions of source and target images. PuzzleMix Kim et al. (2020a) optimally redistributes image patches guided by saliency and local statistics, and Co-Mixup Kim et al. (2021) extends these ideas by simultaneously mixing multiple images with supermodular diversity constraints. SAMix Li et al. (2021) decomposed objectives for mixup generation as local emphasized and global constrained terms in order to learn adaptive mixup mechanism at both class and instance level. SalfMix Choi et al. (2021) transferred a salient region of the image, determined by a saliency map, onto a less salient area within the same image to create a self-mixed training sample. GuidedMixup Kang & Kim (2023) paired images focusing on critical local features within each image via spectral residual.

Most existing salient-based methods focus on extracting salient region from source image and paste it into target image. While, adversarial mixup methods used entire fractal image with high computational resources. In contrast, we propose a ($S^2$-FracMix), first self-saliency method with unique blending approach to overcome self-similarity blending issue while maintain low computational cost in previous methods.

## 3 THE PROPOSED $S^2$-FRACMIX

### 3.1 OVERVIEW

The motivation behind $S^2$-FracMix is to directly encode self-contained multi-scale saliency-guided augmentation. Inspired by Co-Mixup Kim et al. (2020a) and GuidedMixup Kang & Kim (2023) we preserve object saliency while promoting structural diversity. Unlike prior works, we take a direct approach for the detection of gradient-based salient regions via single-pass saliency map Zhang et al. (2020), which is then used to guide patch extractions at various scales. These patches are transformed using rotation and blurring and mixed with the original image at non-salient random positions. Thus $S^2$-FracMix explicitly encodes scale-invariant representation learning while preserving semantic integrity. As shown in Algorithm 1 and illustrated in Figure 3, these multiscale patches are mixed with the original image, ensuring that important information highlighted through saliency is retained, while background or less discriminative areas are altered, to strengthen robust feature learning.

### 3.2 SELF SALIENCY ($S^2$) MIXING

Let $\mathcal{D} = \{(I_i, y_i)\}_{i=1}^{N}$ represent the training dataset, where $I_i \in \mathbb{R}^{c \times h \times w}$ is an input image with $c$ channels, height $h$, and width $w$, and $y_i$ is its corresponding one-hot encoded label. Self Saliency mixup generates an augmented image-label pair $(\tilde{I}_i, \tilde{y}_i)$ through the following steps. Saliency maps $S_i \in \mathbb{R}^{1 \times h \times w}$ are computed to highlight regions critical to the model's predictions

$$S_i = f(I_i, t), \tag{1}$$

where $f(\cdot)$ is the saliency detection method and $t$ is saliency threshold. The saliency maps guide the selection and transformation of patches. Patches are extracted from the salient region of the input image $I_i$ at $n_p$ scales $\mathcal{P} = \{P_1, P_2, \ldots, P_{n_p}\}$. For a patch $P_k$, the patch dimensions are

$$w_k = \lfloor s_k w \rfloor, \quad h_k = \lfloor s_k h \rfloor \tag{2}$$

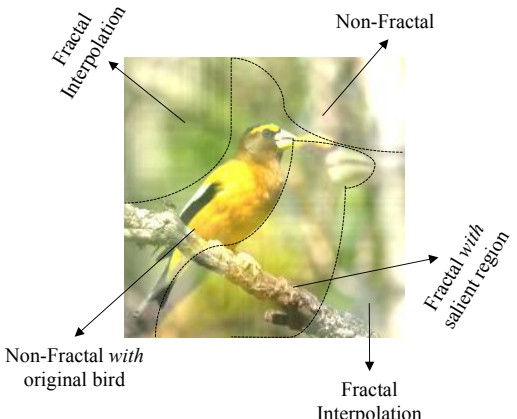

Figure 3: Overview of our proposed $S^2$-FracMix approach. We perform fractal interpolation at multiple scales to generate visual diversity while maintaining semantic consistency.

The top left position $(x_k, y_k)$ for the patch is sampled from salient region $x_k \sim$ Uniform$(0, w - w_k)$, and $y_k \sim$ Uniform$(0, h - h_k)$. The patch is then extracted as: $P_k = I_i[:, x_k : x_k + w_k, y_k : y_k + h_k]$. Each extracted patch $P_k$ is transformed using a pre-defined set of transformations. Let $S_k$ represent the saliency mask for the corresponding patch, defined as: $S_k = S_i[:, x_k : x_k + w_k, y_k : y_k + h_k]$. The transformation $T_k(P_k, S_k)$ is applied as

$$T_k(P_k, S_k) = \text{R}(P_k, \theta) \cdot (1 - S_k) + \text{B}(P_k) \cdot S_k, \tag{3}$$

where, $R(P_k, \theta)$ applies a random rotation $\theta \sim$ Uniform$(-\theta_{\max}, \theta_{\max})$, and $B(P_k)$ applies Gaussian blurring to salient regions in $P_k$. Randomly selected transformed patches are incorporated into non-salient image regions while the others are resized back to the original image dimensions.

$$R_k = \text{Resize}(T_k(P_k, S_k), (h, w)) \tag{4}$$

These resized patches are mixed into the original image using a weighted sum

$$\tilde{I}_i = \alpha I_i + (1 - \alpha) \sum_{k=1}^{n_p} \lambda_k R_k, \tag{5}$$

where $\lambda_k = 1/n_p$ are mixing weights: $\sum_{k=1}^{n_p} \lambda_k = 1$, and $0 \leq \alpha \leq 1$ is a uniform random variable. If patches $P_k$ are taken from different images that have varying class labels, the resulting label will be a weighted combination. Thus, $S^2$ drives learning models to handle a range of spatial transformations without complex mask optimization procedures as used in previous methods such as SaliencyMix Uddin et al. (2020), PuzzleMix Kim et al. (2020a), and Co-Mixup Kim et al. (2021). As a result, our method remains computationally efficient yet highly diverse, synthesizing effective mixing modes that preserves semantic cues.

---

**Algorithm 1:** $S^2$-FracMix Algorithm

---

**Require:** $\mathcal{I}_b = \{I_i, \mathbf{y}_i\}_{i=1}^{b}$: batch of training images with one-hot-encoded labels, $n_p$ patch scales $\mathcal{P}$

**Ensure:** $\tilde{I}_b = \{\tilde{I}_i, \tilde{\mathbf{y}}_i\}_{i=1}^{b}$: Augmented batch with mixed labels

**foreach** *image* $(I_t, y_t) \in I_b$ **do**

    Compute saliency maps: $S_t \leftarrow$ Saliency Map$(I_t)$

    $P_m \leftarrow$ zeros$(w, h)$

    **foreach** *patch* $\mathbf{p}_k(h_k, w_k) \in \mathcal{P}$ **do**

        $x_k \leftarrow$ Uniform$(1, h - h_k)$

        $y_k \leftarrow$ Uniform$(1, w - w_k)$

        Image patch: $P_k \leftarrow I_t[:, x_k : x_k + h_k, y_k : y_k + w_k]$

        Saliency patch: $S_k \leftarrow S_t[:, x_k : x_k + h_k, y_k : y_k + w_k]$

        Fractal blending in $P_k$ using Equation 6

        Transform: $T_k \leftarrow$ R$(P_k, \theta) \cdot (1 - S_k) +$ B$(P_k) \cdot S_k$

        Resize patch: $R_k \leftarrow$ Resize$(T_k, (h, w))$

        Mixed patch: $P_m \leftarrow P_m + \frac{1}{n_k} R_k$

    **end**

    $\alpha \leftarrow$ Uniform$(0, 1)$

    Accumulate into final: $\tilde{I}_t \leftarrow \alpha I_t + (1 - \alpha) P_m$

**end**

$\widetilde{\mathcal{I}}_b \leftarrow \widetilde{\mathcal{I}}_b \cup \{(\widetilde{I}_t, \widetilde{\mathbf{y}}_t)\}$

**return** $\tilde{\mathcal{I}}_b$

---

### 3.3 FRACMIX

In traditional methods, fractals are blended with the whole input image. In contrast, in the current work, we propose fractal blending in salient patches. In self-saliency mixup as discussed above, patches $P_k$ are selected from salient regions. These patches are then blended with self-similarity fractals $F$ to induce structural variations in these patches. Specifically, randomly selected fractal image $F \in \mathcal{F}$ is blended with $P_k$ with a blending factor $\lambda$ as:

$$P_k^f = \lambda F + (1 - \lambda) P_k, \tag{6}$$

The resulting $P_k^f$ is resized and transformed to get $R_k^f$ using Equation 4.

### 3.4 HIGH-LEVEL MIXING OF MULTIPLE MIXING MODES

Most existing approaches employ a *single* mixing strategy throughout training, such as Mixup Guo et al. (2019), CutMix Yun et al. (2019), ResizeMix Qin et al. (2020), PuzzleMix Kim et al. (2020a), and GuidedMixup Kang & Kim (2023). We observe that restricting the model to only one mode of low-level mixing limits the diversity of supervisory signals resulting in performance degradation. Therefore, in this work, we propose to incorporate *multiple low-level modes of mixing* within the training pipeline. Specifically, we mix computationally efficient methods including Mixup Guo et al. (2019), CutMix Yun et al. (2019), and ResizeMix Qin et al. (2020) together at high-level with our proposed $S^2$-FracMix method. For a training instance, one of these methods is randomly selected to encourage complementary regularization effects and expose the model to a richer variety of mixed inputs, ultimately improving robustness and generalization.

## 4 EXPERIMENTS AND RESULTS

We benchmark our proposed $S^2$-FracMix against several recent competitive mixup approaches, including Mixup Zhang et al. (2018), CutMix Yun et al. (2019), ManifoldMix Verma et al. (2019), FMix Harris et al. (2020), ResizeMix Qin et al. (2020), SaliencyMix Uddin et al. (2020), PuzzleMix Kim et al. (2020a), AutoMix Liu et al. (2022b), and AdAutoMix Liu et al. (2022b). Additionally, we report the computational overhead of our method and compare it with timings reported in AdAutoMix Kang & Kim (2023). To demonstrate the generalizability of our method, we conduct experiments from small-scale to large-scale backbones including ResNet18 He et al. (2016), ResNet34 He et al. (2016), ResNet50 He et al. (2016), ResNeXt50 Xie et al. (2017),

Table 1: Top-1 performance (%)↑ of mixup methods on CIFAR-100, Tiny-ImageNet and ImageNet-1K. The results of previous mixup SOTA methods are taken from AdAutoMix Qin et al. (2024). Res18, ResXt50 CNext-T and Res34 refers to ResNet18, ResNeXt50, ConvNeXt-T and ResNet34. Also, ViT-B results are taken from Bai et al. (2022).

| Method | CIFAR-100 | | CIFAR-100 | | Tiny-ImageNet | | ImageNet-1K | | | |
| | Res18 | ResXt50 | Swin-T | CNeXt-T | Res18 | ResXt50 | Res18 | Res34 | Res50 | ViT-B |
|---|---|---|---|---|---|---|---|---|---|---|
| Vanilla | 78.04 | 81.09 | 78.41 | 78.70 | 61.68 | 65.04 | 70.04 | 73.85 | 76.83 | 76.7 |
| MixUp | 79.12 | 82.10 | 76.78 | 81.13 | 63.86 | 66.36 | 69.98 | 73.97 | 77.12 | 80.8 |
| CutMix | 78.17 | 81.67 | 80.64 | 82.46 | 65.53 | 66.47 | 68.95 | 73.58 | 77.17 | 79.9 |
| SaliencyMix | 79.12 | 81.53 | 80.40 | 82.82 | 64.60 | 66.55 | 69.16 | 73.56 | 77.14 | – |
| FMix | 79.69 | 81.90 | 80.72 | 81.79 | 63.47 | 65.08 | 69.96 | 74.08 | 77.19 | – |
| PuzzleMix | 81.13 | 82.85 | 80.33 | 82.29 | 65.81 | 67.83 | 70.12 | 74.26 | 77.54 | – |
| ResizeMix | 80.01 | 81.82 | 80.16 | 82.53 | 63.74 | 65.87 | 69.50 | 73.88 | 77.42 | – |
| AutoMix | 82.04 | 83.64 | 82.67 | 83.30 | 67.33 | 70.72 | 70.50 | 74.52 | 77.91 | – |
| AdAutoMix | 82.32 | 84.22 | 84.33 | 83.54 | 69.19 | 72.89 | 70.86 | 74.82 | 78.04 | – |
| $S^2$-FracMix | 82.74 | 84.91 | 85.35 | 84.41 | 70.38 | 74.27 | 71.37 | 75.34 | 78.54 | 81.2 |

Table 2: Accuracy (%) ↑ of mixup methods on Caltech Birds-200, FGVC-Aircrafts and Stanford-Cars.

| Method | Caltech Birds-200 | | FGVC-Aircrafts | | Stanford-Cars | |
| | ResNet18 | ResNet50 | ResNet18 | ResNeXt50 | ResNet18 | ResNeXt50 |
|---|---|---|---|---|---|---|
| Vanilla | 77.68 | 82.38 | 80.23 | 85.10 | 86.32 | 90.15 |
| MixUp | 78.39 | 82.98 | 79.52 | 85.18 | 86.27 | 90.81 |
| CutMix | 78.40 | 83.17 | 78.84 | 84.55 | 87.48 | 91.22 |
| ManifoldMix | 79.76 | 83.76 | 80.68 | 86.60 | 85.88 | 90.20 |
| SaliencyMix | 77.95 | 81.71 | 80.02 | 84.31 | 86.48 | 90.60 |
| FMix | 77.28 | 83.34 | 79.36 | 86.23 | 87.55 | 90.90 |
| PuzzleMix | 78.63 | 83.83 | 80.76 | 86.23 | 87.78 | 91.29 |
| ResizeMix | 78.50 | 83.41 | 78.10 | 84.08 | 88.17 | 91.36 |
| AutoMix | 79.87 | 83.88 | 81.37 | 86.72 | 88.89 | 91.38 |
| AdAutoMix | 80.88 | 84.57 | 81.73 | 87.16 | 89.19 | 91.59 |
| $S^2$-FracMix | 81.84 | 85.73 | 82.81 | 88.34 | 90.56 | 92.86 |

transformer-based architectures including `Swin Transformer` Liu et al. (2021) and `ConvNeXt` Liu et al. (2022a), and contrastive method `MoCo v2` Chen et al. (2020) and `SimSiam` Chen & He (2021). All experiments are implemented using open-source OpenMixup.

Note that, we follow the standard evaluation practices and protocols mentioned in AdAutoMix Qin et al. (2024) for fair comparison. Hyperparameter configurations and brief implementation guidelines, with detailed settings provided in Appendix B and dataset statistics are also provided in Appendix A. Finally, we show that our proposed $S^2$-FracMix not only improves classification performance across both general- and fine-grained tasks, but also enhances robustness to distributional shifts, such as background corruption Hendrycks et al. (2020), data scarcity, transfer learning, calibration, contrastive learning methods, object detection while maintaining minimal computational overhead. Further results are provided in Appendix D, calibration in Appendix D.1, few-shot learning in Appendix D.2, object detection Appendix D.3 and corrupted dataset Appendix D.4.

## 4.1 GENERAL CLASSIFICATION

We compare the performance of $S^2$-FracMix in Table 1, our approach achieves SOTA performance on CNNs and ViTs, consistently outperforming existing augmentation strategies such as AdAutoMix Qin et al. (2024), AutoMix, ResizeMix, and PuzzleMix. Notably, $S^2$-FracMix surpasses AdAutoMix, the existing best performing method Qin et al. (2024), by approximately **0.42%** and **0.69%** in Top-1 accuracy on CIFAR-100. The trend is similar across different backbones from small-scale to large-scale backbones. In terms of Tiny-ImageNet and ImageNet-1K, the improvement gap is even more pronounced, underlining $S^2$-FracMix capacity to capture rich discriminative features. These results demonstrate that $S^2$-FracMix not only addresses the challenges inherent in CIFAR-100, Tiny-ImageNet, and ImageNet-1K but also establishes a new SOTA among mixup based augmentation methods for enhancing generalization performance.

Table 3: Top-1 accuracy (%) of ResNet-50 (R50) and Vision Transformer (ViT) backbones on CUB-200 and Stanford Cars datasets.

| Backbone | Dataset | Vanilla | MixUp | CutMix | PuzzleMix | AutoMix | AdAutoMix | $S^2$-FracMix |
|---|---|---|---|---|---|---|---|---|
| R50 | CUB-200 | 81.76 | 82.79 | 81.67 | 82.59 | 82.93 | 83.36 | **84.42** |
| R50 | Stanford Cars | 88.88 | 89.45 | 88.99 | 89.37 | 88.71 | 89.65 | **90.85** |
| ViT-B | CUB-200 | 88.0 | 88.75 | 87.76 | 88.23 | 88.91 | 88.76 | **89.84** |
| ViT-B | Stanford Cars | 91.31 | 91.36 | 91.53 | 91.83 | 92.51 | 91.38 | **92.86** |

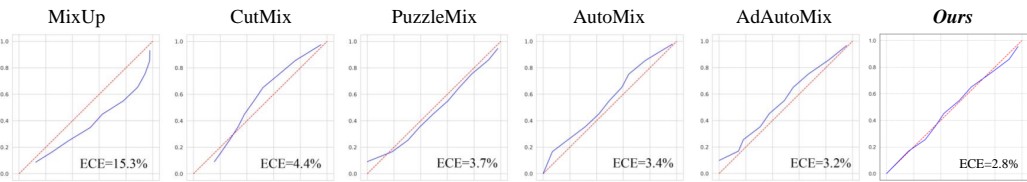

Figure 4: Calibration plots of $S^2$-FracMix using `ResNet18`. Best viewed in Appendix D.1.

## 4.2 FINED-GRAINED VISUAL CLASSIFICATION

In fine-grained classification, we follow the same training protocols established in AdAutoMix Qin et al. (2024) and also previous results are taken from same paper. As reported in Table 2, $S^2$-FracMix consistently achieves the highest Top-1 accuracy across different architectures and datasets. On Caltech Birds-200, $S^2$-FracMix improves over the AdAutoMix by **+0.96%** on `ResNet-18` and **+1.16%** on `ResNet-50`. On FGVC-Aircrafts, it achieves gains of **+1.08%** and **+1.18%** on `ResNet-18` and `ResNeXt-50`, respectively. On Stanford-Cars, improvements of **+1.37%** and **+1.27%** are observed with `ResNet-18` and `ResNeXt-50`. These results demonstrate the general effectiveness of $S^2$-FracMix across fine-grained visual categorization tasks.

## 4.3 TRANSFER LEARNING

Transfer learning, enables efficient adaptation of large-scale models using limited computational resources. We further evaluate the transferability of features learned by $S^2$-FracMix on downstream classification tasks, as presented in Table 3. We utilized two pre-trained deep models including `ResNet-50` and `ViT-B`. Both models are pretrained on ImageNet-1K and fine-tuned on Caltech Birds-200 and Stanford-Cars for classification using $S^2$-FracMix. Compared to AdAutoMix Qin et al. (2024), the strongest existing method, $S^2$-FracMix achieves consistent gains. In Table 3 on Caltech Birds-200, $S^2$-FracMix reaches a Top-1 performances of **84.42%, 89.84%**, outperforming the AdAutoMix by **1.06%, 1.08%**. On the Stanford-Cars, it achieves **90.85%, 92.86%**, exceeding the baseline by **1.20%, 1.48%**. These results demonstrate that $S^2$-FracMix improves fine-tuning performance over baseline and recent SOTA methods across different datasets.

## 4.4 SELF-SUPERVISED LEARNING

A key step in self-supervised learning involves generating two distinct views of an image via data augmentations. $S^2$-FracMix enhances data diversity by introducing more challenging views. In this section, we evaluate the effectiveness of $S^2$-FracMix during the pre-training phase of MoCo v2 Chen et al. (2020) and SimSiam Chen & He (2021). As shown in Table 4, Compared to recent DiffuseMix Islam et al. (2024a), MoCo v2, $S^2$-FracMix achieves a **+2.61%** on Flower102, **+5.09%** on Stanford Cars, and **+3.43%** on Aircraft. Similarly, under SimSiam, $S^2$-FracMix improves performance by **+3.07%** on Flower102, **+3.10%** on Stanford Cars, and **+0.71%** on Aircraft. These results demonstrate that $S^2$-FracMix significantly enhances self-supervised learning, especially on challenging datasets.

Table 4: Top-1 accuracy (%) on Flower102, Stanford-Cars, and Aircraft datasets.

| Method | Flower102 | Stanford Cars | Aircraft |
|---|---|---|---|
| MoCo v2 | 80.31 | 40.82 | 51.36 |
| DiffuseMix | 82.15 | 41.73 | 53.28 |
| $S^2$-FracMix | **84.76** | **46.82** | **56.71** |
| SimSiam | 86.93 | 48.34 | 40.37 |
| DiffuseMix | 89.24 | 49.17 | 42.63 |
| $S^2$-FracMix | **92.31** | **52.27** | **43.34** |

### 4.5 CALIBRATION

Deep Neural Networks often exhibit overconfidence in their predictions during image classification tasks, which can lead to poor calibration. To quantitatively assess calibration performance, we measure the Expected Calibration Error (ECE) across different mixup methods on the CIFAR-100 dataset. Previous figures are taken from AdAutoMix Qin et al. (2024), as illustrated in Figure 4, Our proposed $S^2$-FracMix attains the lowest ECE **2.8%** surpassing recent SOTA methods and second best method is AdAutoMix Qin et al. (2024). More comparison is provided in calibration in Appendix D.1.

### 4.6 ROBUSTNESS

Following the same protocols as used by AdAutoMix Qin et al. (2024), we carried out robustness evaluation experiments under common corruptions on CIFAR100-C Hendrycks & Dietterich (2019) dataset as shown in Table 5. We compared our $S^2$-FracMix with widely used mixup approaches, including CutMix, FMix, PuzzleMix, AutoMix, and AdAutoMix Qin et al. (2024). As shown in Table 5, $S^2$-FracMix demonstrated the best performance on both clean and corrupted samples, achieving relative gains of **1.19%** and **2.4%** in classification accuracy over AdAutoMix. The robustness improvement of **3.14%** is achieved compared to AdAutoMix.

Table 5: Top-1 accuracy and FGSM error of `ResNet-18` with other methods.

| Method | Clean Acc(%)↑ | Corruption Acc(%)↑ | FGSM Error(%)↓ |
|---|---|---|---|
| CutMix | 79.45 | 46.66 | 88.24 |
| FMix | 78.91 | 50.58 | 88.35 |
| PuzzleMix | 79.96 | 51.04 | 80.52 |
| AutoMix | 80.02 | 50.75 | 82.67 |
| AdAutoMix | 81.55 | 51.44 | 75.66 |
| $S^2$-**FracMix** | **82.74** | **53.84** | **72.52** |

## 5 ABLATION AND ANALYSIS STUDY OF $S^2$-FRACMIX

**Inclusion of Simple Modes** We conduct multiple ablation studies to validate the impact of our proposed method $S^2$-FracMix. Table 6 present the results of `ResNet18` and `ResNet50`. We start with our $S^2$-FracMix, which offers two main improvements: i) it generates multi-scale features. ii) saliency-driven patch transformations in a more principled and diverse manner. The introduction of the $S^2$-FracMix leads to a significant gain of 3.69% and 1.13% in terms of performance, highlighting the impact of self-mixing compared to individual performance of each mode $M_m$, $M_c$ and $M_r$.

Here, $M_m$ denotes Mixup Guo et al. (2019), $M_c$ represents CutMix Yun et al. (2019), $M_r$ presents ResizeMix Qin et al. (2020) and $M_f$ illustrates FMix Harris et al. (2020). **Comparison with FMix** While retaining the same training and implementation details, we replace with another mixing mode "$M_f$+$M_m$+$M_c$+$M_r$" this combination degrades the overall performance. **Exclusion of Simple Modes** In our proposed high-level mixing we do not select methods such as PuzzleMix Kim et al. (2020a), Co-Mixup Kim et al. (2021), and GuidedMixup Kang & Kim (2023) demonstrate good performance but in-

Table 6: Ablation study of different high-level mixing strategies on CIFAR-100. First row indicates baseline.

| CIFAR-100 | | | | Accuracy (%) | |
|---|---|---|---|---|---|
| $S^2$-**FracMix** | $M_m$ | $M_c$ | $M_r$ | **ResNet18** | **ResNeXt50** |
| − | − | − | − | 78.04 | 81.09 |
| ✓ | − | − | − | 81.73 | 82.22 |
| − | ✓ | − | − | 79.12 | 82.10 |
| − | − | ✓ | − | 78.17 | 81.67 |
| − | − | − | ✓ | 80.01 | 81.82 |
| ✓ | ✓ | ✓ | - | 82.24 | 82.32 |
| ✓ | ✓ | - | ✓ | 82.46 | 82.89 |
| ✓ | - | ✓ | ✓ | 82.58 | 83.52 |
| ✓ | ✓ | ✓ | ✓ | **82.74** | **84.91** |
| $M_f$ | ✓ | ✓ | ✓ | 80.24 | 82.27 |

cur high computational overhead. As mentioned in Table 6, the best combination is "$S^2$-FracMix+$M_m$+$M_c$+$M_r$" and the reason behind is $M_m$ complements $S^2$ in terms of global inter-image variations, while $M_c$ introduces local inter-image diversity. $M_r$ introduces down-scaled inter-image variations which were missing in the other modes. Thus, the selected set of modes complement each other to generate a diverse set of augmentations. We have included more comprehensive ablation studies in Appendix 13, Appendix 14, Appendix 15, Appendix 16 and Appendix 17.

**Motivation behind High-level Mixing** Four crucial objectives of the current augmentation methods include: *scale-invariance*, *inter-image diversity*, *spatial variability*, and *resolution robustness*. Previous methods address these challenges in isolation. Our objective is to propose a unified high-level mixing framework that jointly tackles all four objectives to achieve SOTA performance while maintaining low computational overhead. In a different ablation study, we replaced $S^2$-FracMix

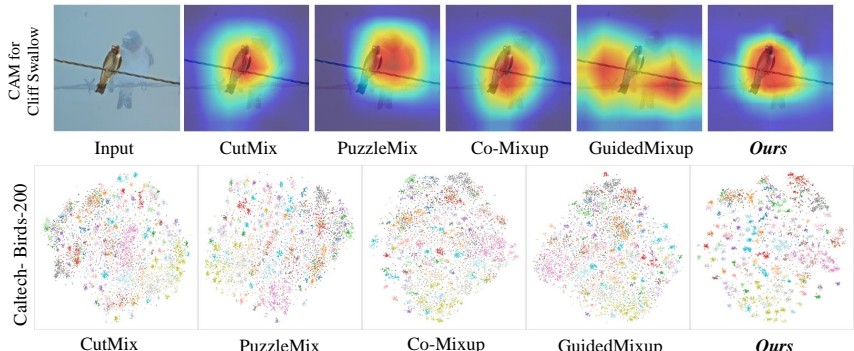

Figure 6: Grad-CAM Selvaraju et al. (2020) visualization on augmented images and T-SNE visualization of `ResNet18` trained from scratch. Detailed figures are provided in Appendix E.1 and E.2.

with $M_f$ as used by Liu et al. (2025) while keeping all other settings the same. We observe reduced performance on two backbones including `ResNet-18` and `ResNeXt-50` in Table 6.

**Hyperparameters Ablation** In the $S^2$-FracMix, there are two main hyperparameters namely saliency threshold $t$ and $\lambda$. In order to achieve good performance both parameters should be properly configured. Firstly, we train the `ResNet18` for 200 epochs via our $S^2$-FracMix. The performance of `ResNet18` with $t$=0.5 is shown in Figure 5 (a). In addition, the fractal mixing in a extracted patch and non-salient region gives performance at $\lambda = 0.20$. However, by increasing the $\lambda = 0.50$, we observe that the classification accuracy is slightly decreased 82.22% and when we set $t$=0.9 the performance degrades to 82.2% which implies that these two parameters are capable of controlling the performance of the $S^2$-FracMix.

**Object Localization** Next, we visually analyze the model trained with $S^2$-FracMix and SOTA methods. As evidenced in Figure 6 (top row), our proposed $S^2$-FracMix produces a contiguous, high-intensity CAM region that consistently highlights the main region, indicating stronger object retention and clearer attention boundaries. More discussion is mentioned in Appendix E.1.

**Feature Representation** Finally, we compare the trained models by visualizing the feature representation of $S^2$-FracMix and SOTA methods in Figure 6 (bottom row). Closely observing t-SNE Van der Maaten & Hinton (2008), it can be seen that images of the same class cluster

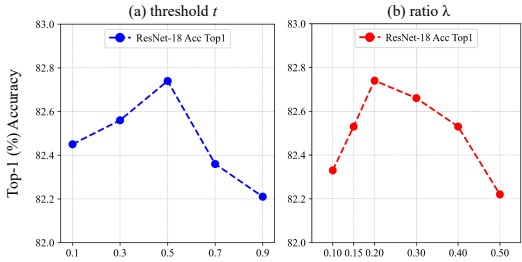

Figure 5: Ablation of hyperparameters $t$ threshold and $\lambda$ for fractal mixing of $S^2$-FracMix on CIFAR100.

together representing better learning. Noticeably, $S^2$-FracMix exhibits distinct and more cohesive clusters with well-defined margins between classes, suggesting that the network consistently learns discriminative features specific to each class. More discussion is mentioned in Appendix E.2.

## 6 CONCLUSION

We introduce $S^2$-FracMix method to improve the performance and generalization of deep learning models. In the proposed $S^2$ mixing, patches of varying sizes are extracted from an input image while utilizing the saliency information. Different transformations are applied on these patches and seamlessly integrated back into the same image. In the proposed *FracMix*, self-similarity fractals are also blended into these salient patches. In this way, training images contain fractal and non-fractal components at the same time, which improves over the previous work. In addition, we also propose high-level mixing of multiple low-level mixing modes to enhance diversity among the augmented samples. Experiments are performed on coarse and fine-grained classification, robustness against corruption, few-shot learning, and transfer learning. The proposed $S^2$-FracMix has demonstrated improved results compared to the existing state-of-the-art methods. The limitation section is provided in Appendix F.

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
