OVERVIEW OF THE APPENDICES

This supplementary material provides additional details that complement the main paper.

- **Section A:** Details of the dataset used for training and testing for our study.

- **Section B:** Details on experimental settings, hyperparameters, and configurations.

- **Section C:** Explanation remaining literature review of data augmentation approaches.

- **Section D:** Further results on few-shot learning, calibration, object detection and corrupted datasets.

- **Section E:** Analysis on MoAE for feature representation and gradcam visulization.

- **Section F:** Mention the limitation of our proposed $S^2$-FracMix work.

- **Section G:** discuss the theoretical analysis of proposed $S^2$-FracMix method.

- **Section H:** mention the detail of saliency-guided patch extraction of the proposed $S^2$-FracMix method.

- **Section I:** experimental detail of self-supervised learning backbones.

## A    DATASET INFORMATION

We provide a concise overview of the datasets used in our study followed by previous SOTA mixup methods Guo et al. (2019); Qin et al. (2024); Yun et al. (2019); Islam et al. (2024a).

**CIFAR-100** Krizhevsky (2009) is widely popular benchmark dataset to evaluate computer vision models. It comprises of $50,000$ training images and $10,000$ test images, each with a resolution of $32 \times 32$. The images are evenly distributed across $100$ distinct classes. These images are evenly distributed across $100$ fine-grained classes, with each class containing 600 images 500 for training and 100 for testing.

**Tiny-ImageNet** Chrabaszcz et al. (2017) is a compact version of the ImageNet dataset Deng et al. (2009). It consists of 200 classes, each containing 500 training images, 50 validation images, and 50 test images. Each image is resized to a fixed resolution of $64 \times 64$ pixels, making the dataset computationally efficient for training and evaluation.

**ImageNet-1K** Deng et al. (2009) is a large-scale image classification benchmark consisting of $1.28$ million training images distributed across $1,000$ classes. Each class contains around $1,300$ images and the dataset also includes a validation set of $50,000$ images. The image resolutions are variable but generally exceed $256 \times 256$ pixels, with resizing applied during preprocessing.

**CUB-200-2011** Wah et al. (2011) is a fine-grained dataset specifically designed for bird species recognition. It contains a total of $11,788$ images covering 200 distinct bird species. Each class has approximately 60 images, although the number of images per class can vary slightly. The dataset is split into $5,994$ training images and $5,794$ testing images, supporting both supervised learning and fine-grained recognition tasks.

**FGVC-Aircraft** Maji et al. (2013) is a fine-grained dataset designed for visual categorization of aircraft models. It comprises $10,000$ images spanning 100 different aircraft classes, where each class corresponds to a specific aircraft model variant (e.g., Boeing 747-400, Airbus A320-200). The dataset is split into training $6,667$ images, validation $1,333$ images, and test $2,000$ images.

**Stanford-Cars** Krause et al. (2013) is a fine-grained dataset consisting of $16,185$ images of cars, split into $8,144$ training images and $8,041$ test images. The dataset includes 196 car classes, each corresponding to a specific make, model, and year. All images are high-resolution, taken from real-world scenarios and curated from car enthusiast websites. The dataset is intended for fine-grained classification, where subtle visual distinctions between very similar car types must be learned.

## B  Implementation Details

We followed the same protocols and configurations mentioned in AdAutoMix Qin et al. (2024) for the fair comparison with SOTA mixup methods. We followed OpenMixup Li et al. (2022) for implementation and experiments.

**CIFAR-100.** We apply basic data augmentations consisting of Random Flip and Random Crop with 4-pixel padding for 32×32 images. For ResNet18 and ResNeXt50, the training setup includes the SGD optimizer (momentum = 0.9, weight decay = 0.0001), a batch size of 100, and 800 training epochs. The initial learning rate is set to 0.1 and adjusted via a cosine scheduler.

**ImageNet-1K.** We adopt a PyTorch-style training configuration on ImageNet-1K, training the model for 100 epochs using the SGD optimizer with a batch size of 256. The initial learning rate is set to 0.1, with a weight decay of 0.0001 and a momentum of 0.9.

**CUB-200, FGVC-Aircraft, and Stanford-Cars.** For CUB-200, FGVC-Aircraft, and Stanford-Cars datasets, we initialize the models using the official PyTorch pre-trained weights on ImageNet-1K. Training is conducted with the SGD optimizer (momentum = 0.9, weight decay = 0.0005), a batch size of 16, and for 200 epochs. The learning rate starts at 0.001 and is adjusted dynamically using a cosine scheduler. The hyperparameters $\lambda$ and $t$ are set to 0.2 and 0.5, respectively.

**Self-Supervised Learning.** We adopt a PyTorch-style training configuration on ImageNet-1K, training the model for 100 epochs using the SGD optimizer with a batch size of 256. The initial learning rate is set to 0.1, with a weight decay of 0.0001 and a momentum of 0.9.

## C  Related Work

### C.1  Data Augmentation

Researchers that developed data augmentation methods aims to improve the performance and generalization of machine learning models Bishop & Nasrabadi (2006); LEE et al. (2020). In the context of computer vision, simple transformations such as random cropping, horizontal flipping, color jittering, and rotation Krizhevsky et al. (2012); Han et al. (2022a) have been standard practice for decades, automatic color enhancement methods Cubuk et al. (2018; 2020); Hataya et al. (2020); Li et al. (2020); Suzuki (2022) to changed original ones. Early efforts in this domain demonstrated that small perturbations of input images could help models learn invariances and reduce overfitting, providing consistent gains across tasks such as image classification Qin et al. (2024); Goodfellow et al. (2014).

### C.2  Generative Mixup Methods

Recently, several generative mixup methods have also been proposed, such as DiffuseMix Islam et al. (2024a), DiffCoRe-Mix Islam & AKHTAR and DiffMix Wang et al. (2024). These methods generate or edit images via contextual prompts to add meaningful features Trabucco et al. (2024); Wang & Chen (2024). Although these methods have shown impressive gains, these techniques may produce out-of-distribution and unfaithful images. It may become particularly important if a pre-trained generative model has not observed the training data distribution to be augmented Zang et al. (2024). Another issue with generative mixup methods is

Table 7: Top-1 accuracy on Flower102 with 10 images per class using `PreActResNet-18` for 300 epochs, from scratch.

| Method | Valid. Acc (%) | Test Acc (%) |
|---|---|---|
| Vanilla | 53.43 | 46.22 |
| Mixup | 60.59 | 52.84 |
| CutMix | 51.96 | 46.37 |
| SaliencyMix | 51.96 | 46.33 |
| PuzzleMix | 61.47 | 54.71 |
| Co-Mixup | 60.49 | 53.08 |
| Guided-AP | 55.49 | 45.94 |
| **Ours** | **63.73** | **56.03** |

their significant computational overhead due to generating new images Wang et al. (2024); Islam et al. (2024b).

In contrast to the generative mixup methods, we consider this work without requiring pre-trained models on the same or overlapping knowledge base. We also aim to design a probabilistic method with reduced computational overhead without affecting the generalization of the neural network. For

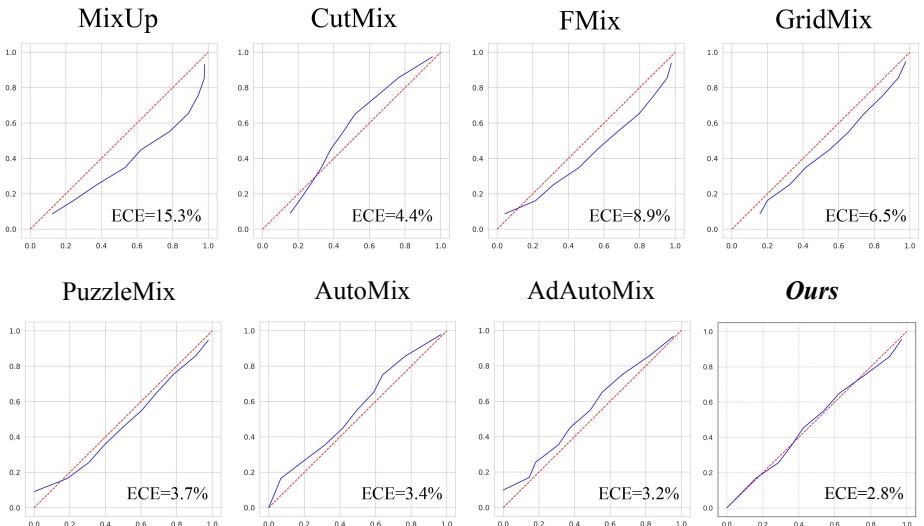

Figure 7: Calibration plots of $S^2$-FracMix on CIFAR100 using `ResNet18`.

this purpose, we propose $S^2$-FracMix, an approach that employs multi-scale and self-mixing method (see Figure 1).

# D  EXPERIMENT AND RESULTS

## D.1  CALIBRATION

The calibration graphs of all recent data augmentation methods are provided in Figure 7. Previous graphs are taken from AdAutoMix Qin et al. (2024) method.

## D.2  COMPARISON ON FEW-SHOT LEARNING

We compare the performance of different algorithms under limited data settings. We limit the number of images in each class to only 10%, 20%, and 50% of the original CIFAR-100 dataset. We evaluate the accuracy of `WideResNet-28-10` trained with various SOTA strategies. As shown in Table 9, $S^2$-FracMix shows excellent gains **17.35%**, **13.8%**, **9.18%** compare to baseline.

$S^2$-FracMix demonstrate superior performance gains when training data is more limited. We also benchmark all compared methods on the Flower102 dataset, which contains just 10 training images per class as shown in Table 7. Compared to Guided-AP Kang & Kim (2023), $S^2$-FracMix obtained significant performance jumps of 8.30% and 10.09 on the validation and test sets. This experiment highlights that $S^2$-FracMix consistently surpasses other mixup techniques in scenarios with limited data availability.

Table 8: Detection performance (mAP %) on SSD and Faster R-CNN.

| Backbone | SSD | Faster R-CNN |
|---|---|---|
| ResNet-50 | 76.7 | 75.6 |
| Mixup | 76.6 | 73.9 |
| Cutout | 76.8 | 75.0 |
| CutMix | 77.6 | 76.7 |
| $S^2$-**FracMix** | **80.26** | **79.48** |

## D.3  OBJECT DETECTION

We evaluate two object detection frameworks including SSD Liu et al. (2016) and Faster R-CNN Girshick (2015) on the Pascal VOC Everingham et al. (2010) benchmark. Note that, we followed protocols of CutMix Yun et al. (2019) while both experts employed VGG-16 as the backbone, we replace it with ResNet-50, initialized with ImageNet pre-trained weights. Fine-tuning is conducted on the combined VOC 2007 and 2012 trainval sets (VOC07+12), and performance is measured using

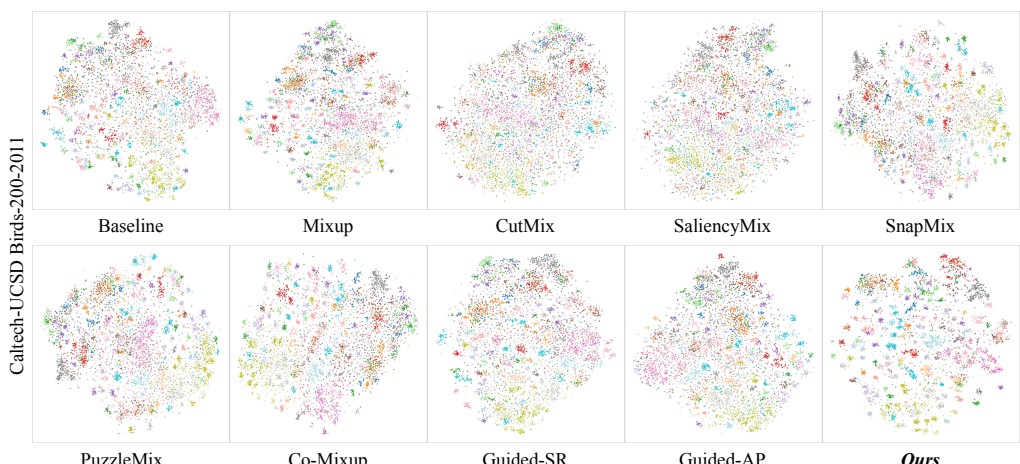

Figure 8: t-SNE visualization on `ResNet-18` trained from *scratch* on the Caltech Birds-200 dataset. The $S^2$-FracMix embeddings show well-defined boundaries with minimal overlap across categories, indicating its effectiveness in preserving fine-grained details and enhancing inter-class separability.

mean Average Precision (mAP) on the VOC 2007 test set. Our fine-tuning follows the protocols of the original works Liu et al. (2016), with further implementation details. The results are provided in Table 8.

### D.4 COMPARISON ON CORRUPTED DATASETS

We evaluate the robustness of our method on the ImageNet dataset corrupted with Gaussian noise and random replacement following Kim et al. (2021). On clean ImageNet dataset, compared to *baseline* the Top-1 and Top-5 error rate is reduced by **2.47%** and **2.00%** (Table 10). For the corrupted dataset, $S^2$-FracMix delivers an improvement of **5.28%** on Gaussian corruption and **3.81%** on random replacement over baseline. This experiment highlights $S^2$-FracMix capability to handle corrupted datasets.

## E ANALYSIS OF $S^2$-FRACMIX

### E.1 FEATURE REPRESENTATION

The feature representation of each method trained with specific data augmentation approach is mentioned in E.1.

### E.2 GRAD-CAM VISUALIZATION

$S^2$-FracMix improves the ability to recognize under partial occlusion. As illustrated in the top row of Figure 11, we selectively obscure regions of the input. In contrast to conventional methods Kang & Kim (2023); Yun et al. (2019), $S^2$-FracMix consistently

Table 9: Accuracy comparison of different methods under varying shots of training data per class on CIFAR-100, using `WideResNet-28-10` from scratch.

| Method | The number of images per class (%) | | |
|---|---|---|---|
| | 50 (10%) | 100 (20%) | 250 (50%) |
| Vanilla | 40.10 | 55.56 | 70.16 |
| Input | 49.44 | 61.74 | 74.31 |
| Manifold | 47.94 | 62.25 | 75.00 |
| CutMix | 42.81 | 60.14 | 74.94 |
| Guided-SR | 50.60 | 63.99 | 75.57 |
| PuzzleMix | 50.13 | 63.99 | 76.31 |
| Co-Mixup | 50.50 | 64.47 | 75.43 |
| Guided-AP | 54.25 | 66.22 | 76.70 |
| **Ours** | **57.45** | **69.36** | **79.34** |

attends to the salient portions of the target object, even when half of it is occluded. Similarly, in the middle row of Figure 11, where two images are mixed, we evaluate whether the trained model can correctly recognize the object. Several models fail to achieve accurate recognition. Likewise, in the bottom row of Figure 11, our proposed method clearly identifies the object despite limited visible cues. This enhanced robustness to partial observations underscores the effectiveness of the proposed method.

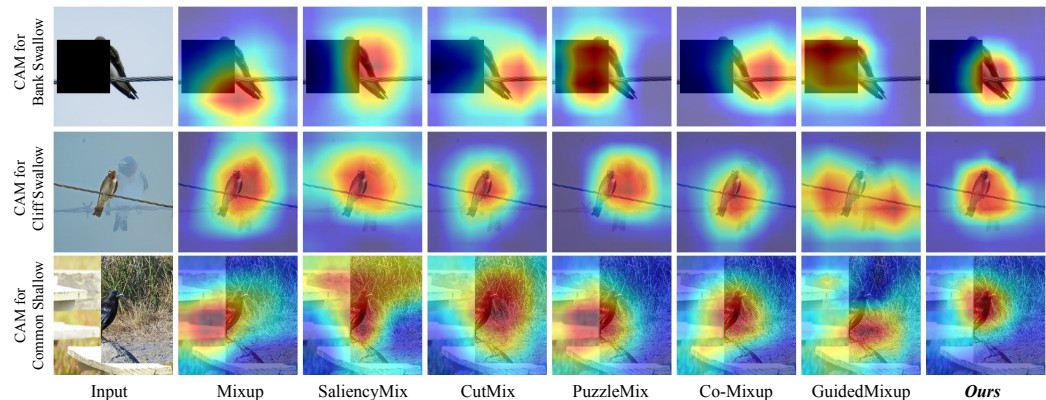

Figure 9: Grad-CAM Selvaraju et al. (2020) visualization on un-augmented and augmented images DeVries & Taylor (2017); Guo et al. (2019); Yun et al. (2019). $S^2$-FracMix effectively identifying target objects even under severe occlusions or mixing scenarios.

### E.3 COMPUTATIONAL OVERHEAD COMPARISON

Our method achieves lower computational overhead primarily because it avoids any *iterative optimization* or *subnetwork* training found in more complex strategies Kim et al. (2020b; 2021). Note that we do not require multiple forward-backward passes to refine masks. Instead, we perform a single saliency estimation and then apply patch transformation. We utilize lightweight operations such as *random resizing*, *blurring*, and *rotation*. In contrast to PuzzleMix Kim et al. (2020a) and Co-Mixup Kim et al. (2020b) which use multiple passes, our method uses a single pass to extract salient regions, resulting in less GPU time usage. Also, we avoid dedicated sub-masks as used in Co-Mixup Kim et al. (2021), reducing extra burden. Consequently, each mini-batch involves only one forward and backward pass, and a small number of random sampling and blending steps. In summary, this compact design is easily scalable to larger datasets while enhancing generalization and robustness of learning models.

We also provide a comparison of the execution times of the saliency detectors used by SaliencyMix, PuzzleMix, AdAutoMix, and $S^2$-FracMix on the CIFAR-100 dataset. As shown in Table 16, our saliency detector accounts for a saving of 27.41 minutes out of the total 162 minutes reduced compared to PuzzleMix. This indicates that the faster saliency detector contributes only 15.06% of the overall speedup. The remaining acceleration arises from the algorithmic design of $S^2$-FracMix, demonstrating that our method is substantially faster than prior approaches beyond the choice of saliency estimator.

We also report the execution-time overhead associated with generating augmented images across the major components of our proposed $S^2$-FracMix in Table 17 for the CIFAR-100 dataset. We observe that $S^2$-FracMix requires only 2.833 minutes to generate all augmented images, while the total training time on a ResNet-50 backbone is 198 minutes. Thus, the augmentation overhead constitutes merely 1.43% of the overall training time.

Table 10: Comparison of error rates on corrupted ImageNet dataset Kim et al. (2021) using `ResNet-50` from scratch.

| Method | Corruption Type | |
|---|---|---|
| | **Gaussian Noise** | **Random Rep.** |
| Vanilla | 29.12 | 41.73 |
| Input | 26.29 | 39.41 |
| CutMix | 27.11 | 46.20 |
| Guided-SR | 26.52 | 38.84 |
| PuzzleMix | 26.11 | 39.23 |
| Co-Mixup | 25.89 | 38.77 |
| Guided-AP | 24.66 | 38.72 |
| **Ours** | **23.84** | **37.92** |

## F  LIMITATIONS

Our proposed $S^2$-FracMix also tackles the important question of **'how many ways to mix?'** while the existing approaches only consider 'one way to mix'. This work presents random selection of mixing modes, which may not fully exploit task-specific or instance-dependent characteristics. While we ensure low computational cost, incorporating gating mechanisms or optimal mixing mode selection would introduce additional augmentation

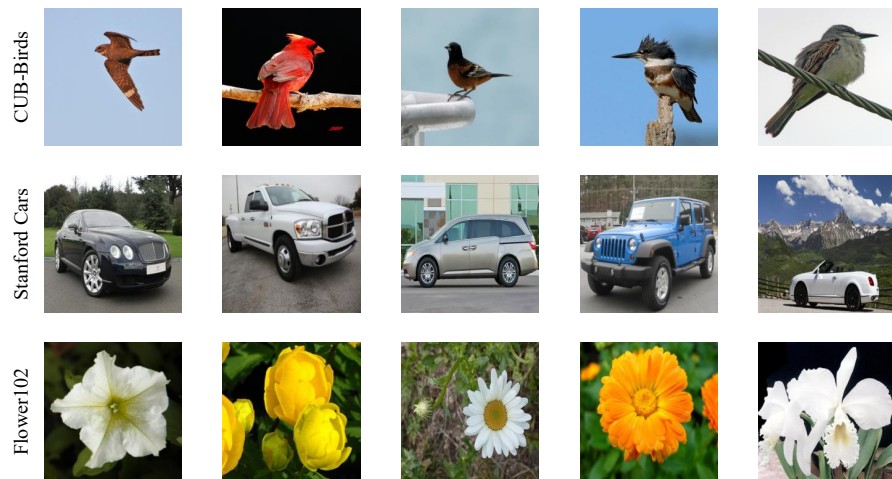

Figure 10: Input batch from the three fine-grained recognition benchmarks used in our experiments including CUB-Birds Wah et al. (2011) Stanford Cars Krause et al. (2013), and Flower102 Nilsback & Zisserman (2008). For each dataset, we randomly pick samples in terms of variability in pose, background, and illumination, highlighting the challenging intra-class variations.

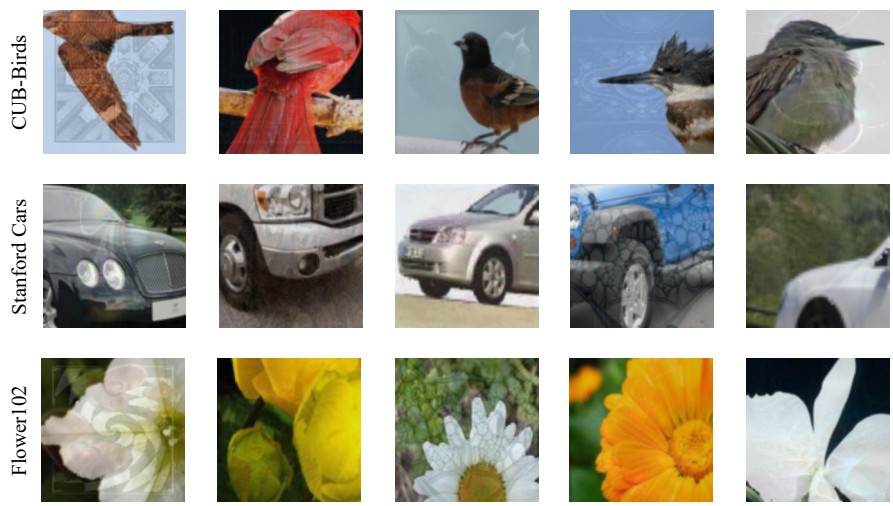

Figure 11: Visualization of patches extracted from salient locations + fractal mixing.

overhead. Moreover, all mixup parameters are manually set rather than learned, favoring reproducibility over adaptability. Our method currently supports only single-modality visual data, limiting its application to video or multimodal settings.

Table 11: Top-1 accuracy (%) of ViT-L 16/224 and ViT-H 14/224 backbones on CUB-200 and Stanford Cars datasets.

| Backbone | Dataset | Vanilla | MixUp | CutMix | $S^2$-FracMix |
|---|---|---|---|---|---|
| ViT-L | CUB-200 | 89.32 | 90.23 | 89.47 | **91.36** |
| | Stanford Cars | 92.41 | 93.0 | 93.53 | **94.64** |
| ViT-H | CUB-200 | 90.46 | 91.15 | 90.67 | **92.25** |
| | Stanford Cars | 93.31 | 93.75 | 94.0 | **95.12** |

Table 12: Top-1 performance (%)↑ of mixup methods on ImageNet-1K. The results of previous mixup SOTA methods are taken from Bai et al. (2022).

| Backbone | Vanilla | MixUp | CutMix | $S^2$-FracMix |
|---|---|---|---|---|
| ViT-B | 76.7 | 80.8 | 79.9 | **81.2** |

Table 13: Ablation of saliency-guided and fractal-based mixing strategies on CIFAR-100 using ResNet18 and ResNeXt50 backbones. "Source A" and "Target B" denote the two images being mixed: *Saliency (A+B)* uses the saliency map of image A to guide mixing with image B, while *Saliency $S^2$ (A+A)* relies on self-saliency from both images. Our fractal-based *FracMix* consistently outperforms the baseline and other saliency variants on both architectures.

| Method | ResNet-18 Top-1 (%) | ResNeXt-50 Top-1 (%) |
|---|---|---|
| Baseline | 78.04 | 81.09 |
| Saliency (A+B) | 79.12 | 81.53 |
| Saliency $S^2$ (A+A) | 79.54 | 81.92 |
| FracMix (ours) | 81.73 | 82.22 |
| $S^2$-**FracMix (ours)** | **82.74** | **84.91** |

# G  THEORETICAL ANALYSIS

Although a full theoretical proof is beyond the scope of this work, we can provide the following intuitive connections that partially explain the empirical success.

**Scale-invariant fractal prior:** Natural images exhibit fractal self-similarity across scales, some works Vidivelli et al. (2023); Hendrycks et al. (2022) had shown that by injecting fractal noise selectively in low-saliency regions could bring improvement, also, we explicitly encourage the network to learn representations that are robust to scale variations, which aligns with the natural image manifold.

**Information bottleneck perspective:** Self-saliency mixing forces the model to reconstruct high-saliency objects using only low-saliency context tokens. This can be viewed as adding a structured auxiliary loss that encourages the minimal sufficient statistic to be distributed across the entire image rather than concentrated in a few dominant regions Tishby et al. (2000); Achille & Soatto (2018), thereby increasing robustness to local perturbations.

| | |
|---|---|
| $x \in [0,1]^{H \times W \times C}$ | Input image; $y$ its label. |
| $\mathcal{S}(x) \in [0,1]^{H \times W}$ | Per-pixel saliency map; threshold at percentile $\tau$ to get salient support $\Omega_\tau = \{p : \mathcal{S}(p) \geq \tau\}$. |
| $S^2$ (self-saliency) | Move a patch from $\Omega_\tau$ to a non-salient location *in the same image* (label preserved). |
| FracMix (localized) | Inside each (re-placed) salient patch apply $(1-\alpha) \cdot \text{patch} + \alpha \cdot F$ where $F$ is a zero-mean fractal field ($\approx 1/f$); edited area $\rho \ll 1$. |
| $A_\theta$ | Augmentation with randomness $\theta$: $x' = A_\theta(x) = T_\theta(x) + \Delta_\theta(x)$ where $T_\theta$ is $S^2$ and $\Delta_\theta$ is the local fractal blend. |

## ASSUMPTIONS

(A1) $\ell \circ f$ is twice differentiable and $L$-smooth near $x$.

(A2) $T_\theta$ (the $S^2$ move) preserves the label $y$.

(A3) $\mathbb{E}[F] = 0$, $\text{Cov}(F) = \Sigma_f$ with spectral density $S(\omega) \propto \|\omega\|^{-\beta}$ ($\beta \approx 1$; a $1/f$-like prior).

(A4) The mask $M_\theta$ is localized to the salient patch; edited area $\rho$ is small (e.g., $8-12\%$).

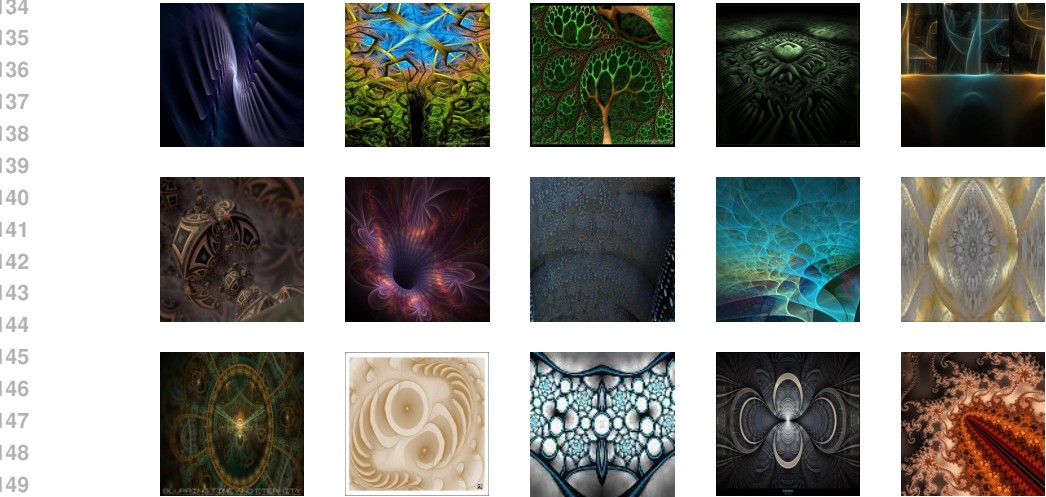

Figure 12: Visualization of self-similarity fractal images taken from DiffuseMix Islam et al. (2024a) fractal dataset. Each subfigure represents a unique complex pattern, demonstrating the diversity and complexity inherently present in fractal geometry.

## KEY DECOMPOSITION

Let $x_T := T_\theta(x)$ and $\Delta_\theta := \alpha M_\theta F$. A second-order expansion around $x_T$ plus $\mathbb{E}[\Delta_\theta] = 0$ gives

$$\mathcal{L}_{\text{VRM}}(f) = \mathbb{E}_{x,y}\mathbb{E}_\theta \,\ell(f(A_\theta(x)),y) \approx \underbrace{\mathbb{E}_{x,y}\mathbb{E}_\theta \,\ell\big(f(T_\theta(x)),y\big)}_{\text{invariance to } S^2 \text{ (context/scale)}} + \underbrace{\tfrac{\alpha^2}{2}\,\mathbb{E}_x \,\text{Tr}\big(\mathcal{H}(T_\theta(x))\,\Sigma_{\text{loc}}(x)\big)}_{\text{saliency-weighted stability penalty}}$$

where $\mathcal{H} := J_f^\top H_\ell J_f$ and $\Sigma_{\text{loc}}(x) := \mathbb{E}[M_\theta \Sigma_f M_\theta^\top]$ concentrates mass on salient pixels.

## SPECTRAL INTUITION

Writing $\mathcal{H} \approx J_f^\top J_f$ for intuition and using Fourier notation $(\hat{\cdot})$,

$$\text{Tr}(\mathcal{H}\,\Sigma_{\text{loc}}) \approx \int S(\omega)\,\|\widehat{J_f}(\omega)\|_F^2\,|\widehat{M_\theta}(\omega)|^2\,d\omega,$$

so the penalty suppresses input-gradient energy *inside* salient patches across a broad (self-similar) band, de-emphasizing brittle micro-textures while preserving shape/parts.

## GENERALIZATION SKETCH OF LOCAL COMPLEXITY

Consider the regularized empirical objective

$$\mathcal{R}_\lambda(f) = \widehat{\mathcal{L}}(f) + \lambda\,\mathbb{E}_x\,\text{Tr}\big(J_f^\top J_f \Sigma_{\text{loc}}(x)\big), \quad \lambda \propto \alpha^2.$$

Standard local Rademacher/PAC-Bayes arguments for Jacobian-controlled classes give (w.p. $\geq 1-\delta$)

$$\mathcal{L}(f) \leq \widehat{\mathcal{L}}(f) + C\sqrt{\frac{\mathbb{E}_x\,\|J_f(x)\|_{\Sigma_{\text{loc}}}^2}{n}} + O\left(\sqrt{\frac{\log(1/\delta)}{n}}\right).$$

Minimizing $\mathcal{R}_\lambda$ shrinks $\mathbb{E}_x\|J_f(x)\|_{\Sigma_{\text{loc}}}^2$ on salient regions, tightening the bound and improving margins/robustness where decisions are made.

## WHY SALIENT-LOCAL FRACTAL ARE BETTER THAN GLOBAL-FRACTAL OR NON-FRACTAL CONTROL?

- **Global fractal** spreads the penalty to background/context and may harm semantics.

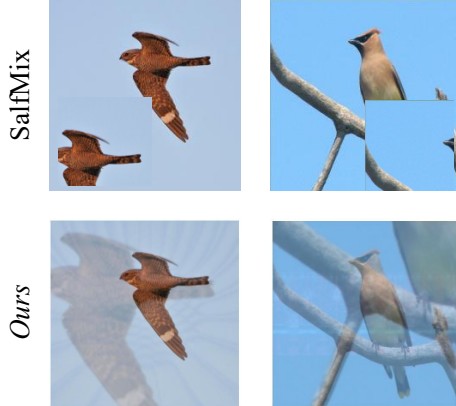

Figure 13: Visual Comparisons of SalMix Choi et al. (2021) and our proposed method $S^2$-FracMix.

- **Non-fractal controls** (Perlin/band-pass) are narrow-band; at equal area/energy they reduce $\|J_f\|_{\Sigma_{\text{loc}}}^2$ less.
- **Salient-local fractal** targets attended regions and covers multiple bands $\Rightarrow$ strongest regularization at fixed budget.

## H  SALIENCY-GUIDED PATCH EXTRACTION DETAILS

For each input image $I \in [0,1]^{H \times W \times C}$, we compute a per-pixel saliency map $\mathcal{S}(I) \in [0,1]^{H \times W}$ using the single-pass gradient-based method of Zhang et al. Zhang et al. (2020). The raw output is normalized to the range $[0,1]$, where higher values indicate spatial regions that contribute more strongly to the model's prediction.

We define the salient support as
$$\Omega_\tau = \{\, p \mid \mathcal{S}(p) \geq \tau \,\},$$
using a fixed threshold $\tau = 0.75$. This threshold selects high-importance spatial locations of the input image and remains constant across all experiments.

From $\Omega_\tau$, we extract salient patches at two predefined scales:
$$P = \{(H/4, W/4),\ (H/2, W/2)\}.$$

Thus, each image yields a small-scale and a medium-scale salient crop. This choice of $n_p = |P| = 2$ was made empirically: using more scales produced only marginal benefit, while using fewer scales reduced performance. Each cropped patch undergoes a set of context-aware transformations, including rotation and optional blurring. The transformed patch $\tilde{P}$ is then resized to the full image resolution $[H, W]$ and blended back into the original image using saliency-weighted mixing as described in Eq. (3) of the main paper:
$$I' = (1 - \alpha \mathcal{S}) \cdot I \ + \ \alpha \mathcal{S} \cdot \tilde{P},$$

Table 14: Comparison on global fractal method: $\tilde{I}_i = I_i + \lambda F$, where $\lambda$ is as proposed by Islam et al. (2024a) vs. $S^2$-FracMix (using local fractal) on Stanford-Cars dataset. These results are taken from Table 14 in Islam et al. (2024a).

| Method | ResNet-50 Top-1 (%) |
| --- | --- |
| Baseline | 85.52 |
| + Fractal (*global*) | 86.73 |
| Mixup Guo et al. (2019) | 88.14 |
| + Fractal (*global*) | 54.25 |
| $S^2$-FracMix (*local*) | **92.78** |

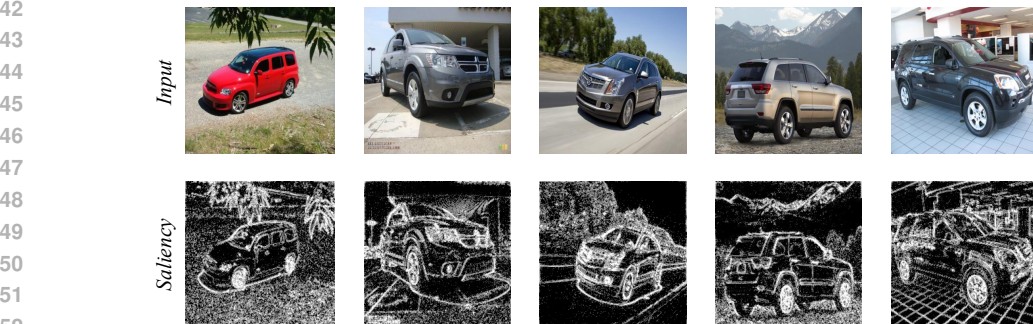

Figure 14: Visualization of failure cases of saliency algorithm used in the proposed $S^2$-FracMix. In some cases, the saliency algorithm includes background region in the saliency map. Despite these failures, our algorithm has achieved excellent performance.

Table 15: Ablation on Stanford-Cars dataset: $S^2$-FracMix using fractal addition to salient region (Eq. 6) vs. $S^2$-FracMix with global fractal addition: $I_i = \lambda F + (1 - \lambda)I_i$. For $S^2$-FracMix without saliency weighting, (Eq. 3) is modified as $T_k(P_k, S_k) = 0.50\,\mathrm{R}(P_k, \theta) + 0.50\,\mathrm{B}(P_k)$.

| Method | ResNet-50 Top-1 (%) |
|---|---|
| Baseline | 85.52 |
| $S^2$-FracMix (*without saliency weighting*) | 91.87 |
| $S^2$-FracMix (*global fractal*) | 92.27 |
| $S^2$-FracMix (*local*) | **92.78** |

where $\alpha$ controls the blending strength.

# I  SSL IMPLEMENTATION DETAILS

We use a standard ResNet-50 encoder for all SSL experiments. The final fully connected layer is removed and replaced by the projection heads used in MoCo v2 and SimSiam DiffuseMix Islam et al. (2024a) and YOCO Han et al. (2022a).

**MoCo v2.**  The projection head is a 2-layer MLP ($2048 \rightarrow 2048 \rightarrow 128$) with ReLU activation. The temperature parameter is set to 0.2. We use a batch size of 256, SGD with momentum 0.9, and a cosine learning-rate schedule for 200 epochs.

**SimSiam.**  The projection head is a 3-layer MLP ($2048 \rightarrow 2048 \rightarrow 2048 \rightarrow 128$) with BN and ReLU. The predictor is a 2-layer MLP ($128 \rightarrow 512 \rightarrow 128$). Training follows the standard SimSiam recipe.

$S^2$-FracMix is applied to one of the two augmented $224x224$ views before encoding, following the same protocol used in prior SSL augmentation studies. No architectural modifications or hyperparameter changes were made beyond integrating our augmentation.

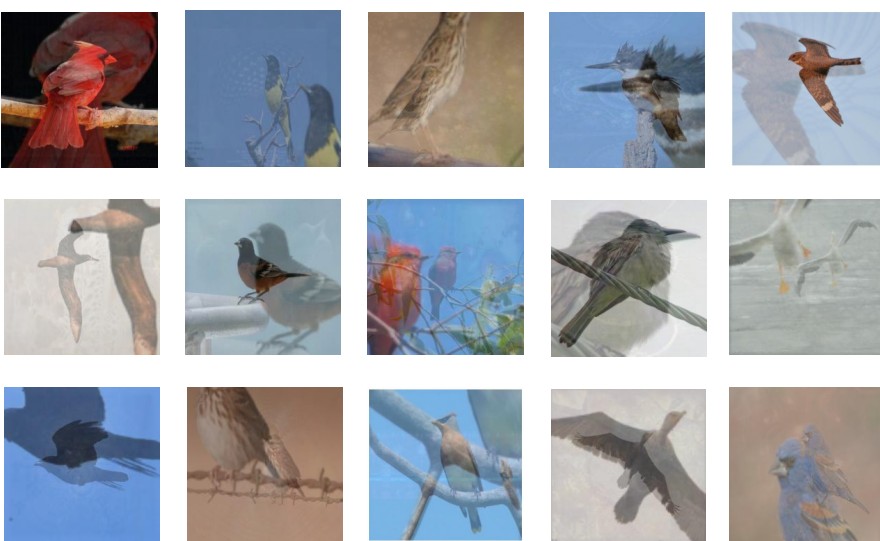

Figure 15: Visualization of our final augmented samples generated from our proposed $S^2$-FracMix method.

Table 16: Execution time comparison of saliency detectors used by SaliencyMix, PuzzleMix, AdAutoMix, and $S^2$-FracMix on CIFAR-100 dataset. Our saliency detector saves 27.41 minutes from the overall saving of 162 minutes of overall training compared to PuzzleMix.

| Method | Saliency Detector Time ($min$) | Overall Training Time ($min$) |
|---|---|---|
| SaliencyMix Uddin et al. (2020) | 0.014 | 198 |
| PuzzleMix Kim et al. (2020a) | 27.91 | 360 |
| AdAutoMix Qin et al. (2024) | 20.35 | 336 |
| $S^2$-FracMix | **0.500** | **198** |

Table 17: $S^2$-FracMix augmented images generation time taken by different components on CIFAR-100 dataset. The overall training time depends on the choice of specific backbone.

| Method | Execution Time ($min$) |
|---|---|
| Saliency Detector | 0.50 |
| Transformations | 0.01 |
| Fractal Blending | 1.00 |
| Multi-mode Mixing | 1.75 |
| Total | **2.833** |