# OpenReview forum: "$S^{2}$-FracMix: Self-Saliency Fractal Mixup"
_ICLR.cc/2026/Conference — ICLR 2026 Conference Withdrawn Submission_

### Official Review · Reviewer_njcS · 2025-10-15

**Soundness:** 2
**Presentation:** 2
**Contribution:** 2
**Rating:** 4
**Confidence:** 4

**Summary:**

S2-FracMix introduces a novel data augmentation framework that integrates self-saliency mixing with fractal texture injection to enhance model robustness against local deformation and texture perturbation. The method is structurally simple and highly generalizable—it requires no additional data, complex optimization, or architecture modification, and can be easily applied to both CNNs and Vision Transformers. By mixing self-salient patches and injecting fractal structures, S2-FracMix preserves semantic consistency while significantly enriching feature diversity and improving generalization. Extensive experiments across multiple datasets and tasks demonstrate consistent and competitive performance, highlighting its strong practical value.

**Strengths:**

1.	S2-FracMix is lightweight and easy to implement. It does not rely on additional data, external generators, or complex optimization procedures. The method can be seamlessly integrated into CNN or Transformer training pipelines with minimal computational overhead, making it highly practical and broadly applicable.
2.	The use of self-saliency ensures that the augmentation process preserves the main semantic content of an image. Meanwhile, the self-mixing of salient patches and fractal injection enrich local variations in scale, shape, and texture. Combined with multi-mode mixing, this produces more diverse and semantically stable samples, significantly improving model generalization and robustness.
3.	The authors evaluate S2-FracMix across multiple benchmark datasets and compare it with various state-of-the-art augmentation methods. It consistently achieves better classification accuracy, robustness, and calibration, and performs well in supervised, semi-supervised, and transfer learning settings — demonstrating stable and reliable performance across tasks.

**Weaknesses:**

1.	The effectiveness of S2-FracMix heavily relies on the accuracy of the saliency map. If salient regions are detected incorrectly, the mixing process may disrupt semantics or produce suboptimal augmentations, leading to unstable performance in noisy or complex data scenarios.
2.	The fractal injection strength, controlled by the parameter λ, must be carefully tuned. Excessive injection may distort local textures or corrupt fine-grained semantics, particularly in detailed classification tasks. Proper parameter selection is essential to balance augmentation strength and semantic fidelity.
3.	A main limitation of the paper is that it does not clearly verify the true source of its performance gains. Although results are strong, the authors attribute improvements mainly to fractal injection without isolating its effect or comparing alternative designs, such as using salient shapes from other images with the current image’s textures. Without such ablations, it remains unclear whether the advantage comes from the fractal patterns themselves or simply from increased structural diversity.

**Questions:**

Please see weakness

---

> ### Author Response · Authors · 2025-11-21
> **Authors’ Response to Weaknesses identified by Reviewer njcS**
>
> First of all, we would like to thank reviewer njcS for taking the time to review our work and constructive feedback.
>
> **(W 1) The effectiveness of S2-FracMix heavily relies on the accuracy of the saliency map. If salient regions are detected incorrectly, the mixing process may disrupt semantics or produce suboptimal augmentations, leading to unstable performance in noisy or complex data scenarios.**
>
> We thank the reviewer for highlighting this important implementation detail. Our objective is to design a label-preserving method that avoids generating unrealistic images while maintaining a low memory footprint. If the saliency detector moderately misidentifies salient regions, the overall performance of our method is largely unaffected. However, if the saliency detector fails severely, it may reduce the effectiveness of this step, but the degradation would remain graceful. A figure showing failure cases of the saliency detector used in our experiments is shown in **Appendix (Fig. 14)**. Despite these failures, S$^2$-FracMix has obtained excellent performance.
>
> **(W 2) The fractal injection strength, controlled by the parameter λ, must be carefully tuned. Excessive injection may distort local textures or corrupt fine-grained semantics, particularly in detailed classification tasks. Proper parameter selection is essential to balance augmentation strength and semantic fidelity.**
>
> We performed the ablation study on λ by varying its values from 0.10, 0.15, 0.20, 0.30,0.40, 0.50 as shown in **Fig. 5 (b).** The performance variation remains between 82.33% to 82.74% (mean 82.47 and variance 0.0625), which is not a very significant variation.  Therefore, the proposed method is not very sensitive to the λ parameter. Please note that the vertical scale in **Fig. 5 (b)** varies from 82.33 to 82.74 with a range of only 0.41% variation. Due to the very small range of vertical axes, sharp changes are visible. The text and the **Fig. 5** (main paper) are modified accordingly.
>
> **(W 3) A main limitation of the paper is that it does not clearly verify the true source of its performance gains. Although results are strong, the authors attribute improvements mainly to fractal injection without isolating its effect or comparing alternative designs, such as using salient shapes from other images with the current image’s textures. Without such ablations, it remains unclear whether the advantage comes from the fractal patterns themselves or simply from increased structural diversity.**
>
> We thank Reviewer njcS for the constructive suggestions. In response, we conducted additional ablation experiments with **self-saliency S² (A+A)**, **cross-saliency (A+B)**, **fractal image mixing (FracMix)**, and **multiple modes of mixing (S$^2$-FracMix)** on CIFAR-100 using two backbones, as reported in **Appendix Tab. 13**. We observe that self-saliency S² (A+A) is more accurate than cross-saliency (A+B). The performance gains of self-saliency over the baseline are 1.50% on ResNet-18 and 0.83% on ResNeXt-50. Adding FracMix further improves accuracy by 2.19% on ResNet-18 and 0.30% on ResNeXt-50. Incorporating multiple modes of mixing yields additional improvements of 1.01% and 2.69% on the two backbones, respectively. These results indicate that fractal image mixing is an important component of our algorithm, but not the only factor driving performance: each proposed step contributes meaningfully to the overall performance gains.

---

> ### Author Response · Authors · 2025-11-25
> **Encouraging Discussion with Reviewer njcS**
>
> Dear Reviewer njcS,
>
> Thank you for your insightful comments, which significantly enhanced the clarity, rigor, and overall quality of our work. We have carefully addressed all your suggestions, and we hope the revised manuscript now meets your expectations. We would be sincerely grateful if you could kindly reconsider your evaluation. Should you have any further questions or require additional clarification, we would be happy to continue the discussion.
>
> Warm regards,
>
> The Authors

---

> ### Comment · Reviewer_njcS · 2025-11-25
> **response to authors**
>
> My concerns have been fully solved

---

> > ### Author Response · Authors · 2025-11-26
> > **Thank you**
> >
> > Thank you very much for taking the time to review our work and for re-evaluating it. We sincerely appreciate your thoughtful questions and your careful assessment of our responses and new experiments. We’re glad that we were able to address all your concerns, and we truly appreciate your decision to increase your rating from 4 to 6.

---

### Official Review · Reviewer_N3st · 2025-10-29

**Soundness:** 3
**Presentation:** 3
**Contribution:** 3
**Rating:** 4
**Confidence:** 4

**Summary:**

This paper proposes a new data augmentation method combining saliency-based mixing and fractal mixing. The effectiveness of the proposed method is verified in 7 datasets with different tasks showing higher performance compared to the exisiting methods.

**Strengths:**

1. This paper firstly incorporates the saliency mixing and fractal mixing to improve DA performance.
2. Intensive experiemts are performed in various datasets and tasks, making the proposed method convincing.
3. The proposed method is very fast while showing the best performance in all the tasks in the paper.

**Weaknesses:**

1. Although the proposed method reveals better performance, the novelty seems limited. That is, saliency, fractal mixing methods already exisit and this method combined them sequentially.

2. Augmented samples are not visualized. Especially, factal mixing results for salient patches are not shown in the paper, which limits the possibility of analysis on augmented representations and the generalization power.

3. The motivation is unclear. That is, the fractal mixing was devised for adversarial robustness, and this paper applied it to salient regions. Why do this combination results in better perforamce to clean images (including adversarial robustness)? What is the underlying mechanism of fractal mixing for salient patches in terms of enhancing generalization performance? One can think that data augmentation should consder the trade off between diversity and fidelity for the augmented data. Such analysis is missing which makes this work seem heuristic.
4. As far as I know. saliency mixing is very fast since it compute the saliency map only for the first image in a mini batch, decide the position to be cropped and attached. This position is applied to all the remaining samples in the batch. How did you measure the computational complexity of it? It should be checked carefully.

**Questions:**

Please see the Weakness and answer the all concerns in it.

---

> ### Author Response · Authors · 2025-11-21
> **Authors’ Response to Weaknesses identified by Reviewer N3st**
>
> We would like to thank the reviewer N3st for taking the time to review our work and constructive feedback.
>
> **(W 1) Although the proposed method reveals better performance, the novelty seems limited. That is, saliency, fractal mixing methods already exisit and this method combined them sequentially.**
>
> The core novelty of our work lies in the targeted multi-scale self-saliency patch extraction and the selective fractal blending strategy.  Particularly, S$^2$-FracMix extracts patches of different sizes from salient regions, applies blurring and rotation, and resizes them to full image dimensions. It integrates fractal blending within the salient patches to enhance robustness. Then blends them back into the original image using saliency-based weights. This integrative design produces a unique augmentation effect that cannot be replicated by merely combining existing techniques.  We introduce a high-level, multi-mode mixing scheme where the augmentation mode is randomly selected for each training instance, improving the model’s generalization. Our method not only improves performance but also introduces adaptive diversity and enhanced robustness beyond prior approaches. Overall, this work establishes a new baseline and represents a meaningful step forward in the field of data augmentation.
>
> **(W 2) Augmented samples are not visualized. Especially, fractal mixing results for salient patches are not shown in the paper, which limits the possibility of analysis on augmented representations and the generalization power.**
>
> We have added these visualizations to the supplementary material. **Fig. 10** shows the input images, **Fig. 11** shows the patches extracted from salient regions with added fractals, and **Fig. 12** presents fractal images that are blended in salient regions. **Fig. 15** shows the final augmented images. We hope these visuals help clarify the behaviour of our method.
>
> **(W 3) The motivation is unclear. That is, the fractal mixing was devised for adversarial robustness, and this paper applied it to salient regions. Why do this combination results in better perforamce to clean images (including adversarial robustness)? What is the underlying mechanism of fractal mixing for ....**
>
> The motivation for blending fractal patterns into salient patches in S$^2$-FracMix is rooted in precisely balancing diversity and fidelity in data augmentation. Previous fractal mixing methods such as **PixMix** [CVPR’22], **IPMix** [NeurIPS’23], **DiffuseMix** [CVPR’24] added global fractal noise for adversarial robustness but risked eclipsing semantic image features, causing a trade-off where robustness improved but fidelity may suffer.
>
> S$^2$-FracMix innovates by first restricting fractal injections to only the saliency-guided regions, guided by the intuition that semantically critical content benefits more from structural complexity. Later on, these fractal blended salient regions are rescaled to full image size and added in the original image.  Experiments show this mechanism enables models to learn more discriminative, scale-invariant features that generalize better to both clean, and adversarially perturbed images, outperforming baselines in both regimes and visualized through **CAM (Fig. 6)** and **t-SNE clustering  (Fig. 6)**. We add a new ablation comparing the performance of global fractal vs local fractal and observed performance degradation when global fractal is added in **Appendix Tab 14**.
>
> **(W 4) As far as I know. saliency mixing is very fast since it compute the saliency map only for the first image in a mini batch, decide the position to be cropped and attached....**
>
> - We thank the reviewer for highlighting this important implementation detail. As noted, if SaliencyMix computes the saliency map only for the first image in each mini-batch and reuses the same pasting location for all remaining samples, it becomes highly efficient but risks occluding critical regions in many images due to the fixed placement. In contrast, subsequent saliency-guided methods (e.g., **TransMix** [CVPR’22], **CoMixup** [ICLR’21], **PuzzleMix** [ICML’20]) generate per-sample mixing masks through independent encoding and additional metrics, achieving higher accuracy but at the cost of substantial computational overhead.
> - Our proposed S$^2$-FracMix offers a more favorable trade-off. It avoids the information loss caused by fixed-location pasting while remaining significantly lighter than per-sample saliency-based methods such as **ResizeMix** [arxiv’20], **PuzzleMix** [ICML’20], **AutoMix** [ECCV’22], and **AdAutoMix** [ICLR’24] (see **Fig. 2** in the main paper). This efficiency primarily stems from eliminating the need to compute saliency maps for every image, made possible by our high-level strategy of mixing across multiple augmentation modes.

---

> ### Author Response · Authors · 2025-11-25
> **Encouraging Discussion of Rebuttal**
>
> Dear Reviewer N3st,
>
> We have thoroughly integrated all of your suggestions and expanded our analyses to address each of your comments in detail. We hope the revised manuscript now meets the level of quality you expect, and we would greatly appreciate it if you could consider these improvements in your final evaluation. If any additional clarification is needed, we would be glad to provide it.
>
> Warm regards,
>
> The Authors

---

### Official Review · Reviewer_Rnnd · 2025-10-30

**Soundness:** 2
**Presentation:** 3
**Contribution:** 2
**Rating:** 4
**Confidence:** 4

**Summary:**

This paper proposes S2-FracMix, a novel data augmentation method that combines two key ideas: (1) Self-Saliency (S2) mixing, which extracts multi-scale salient patches from an image and reinserts them into non-salient regions of the same image after applying transformations (e.g., rotation, blur); and (2) FracMix, which injects self-similar fractal patterns only into those salient patches to enhance structural diversity while preserving semantic consistency. Additionally, the authors adopt a high-level mixing strategy that randomly selects among several augmentation modes (including Mixup, CutMix, ResizeMix, and S2-FracMix) during training to increase regularization diversity. The method is evaluated across a range of tasks including general/fine-grained classification, robustness to corruption, calibration, few-shot learning, transfer learning, and self-supervised learning on datasets such as CIFAR-100, Tiny-ImageNet, ImageNet-1K, CUB-200, and Stanford Cars. The results consistently show S2-FracMix outperforming prior state-of-the-art mixup methods with lower computational overhead.

**Strengths:**

* Innovative design: The idea of intra-image saliency-guided mixing (S2) is conceptually distinct from prior inter-image saliency methods (e.g., PuzzleMix, Co-Mixup), reducing computational cost while maintaining semantic fidelity.
* Targeted fractal augmentation: Restricting fractal blending to salient regions (FracMix) avoids the distribution shift caused by global fractal mixing (e.g., in PixMix or DiffuseMix), which is a thoughtful improvement.
* Strong empirical performance: The method demonstrates consistent gains across diverse tasks and architectures (CNNs and ViTs), including robustness, calibration, and transfer learning, which suggesting broad applicability.
* Efficiency: The paper convincingly shows lower training time compared to heavy saliency-based methods, making it more practical for real-world use.

**Weaknesses:**

* Limited scale of experiments: While the paper claims generalizability, all experiments are conducted on relatively small to medium-scale datasets (e.g., CIFAR-100, Tiny-ImageNet) and architectures (up to ResNet-50, ViT-B). There is no evaluation on truly large-scale settings, such as full ImageNet-21K pretraining, billion-parameter models, or modern large vision-language models, which are increasingly standard in top-tier vision and learning venues like ICLR.
* Absence of large model evaluation: The largest backbone tested is ViT-Base. Given the growing importance of scaling in modern ML, the omission of experiments with larger transformers (e.g., ViT-L, ViT-H) or foundation models weakens the claim of broad applicability.
* Lack of theoretical analysis: The method is presented purely from an empirical and heuristic perspective. There is no theoretical justification for why self-saliency mixing or localized fractal injection should improve generalization or robustness—e.g., no connection to invariance principles, information theory, or optimization dynamics. This limits insight into why the method works and under what conditions it might fail.

**Questions:**

See Weakness

---

> ### Author Response · Authors · 2025-11-21
> **Authors’ Response to Weaknesses identified by Reviewer Rnnd**
>
> First of all, we would like to thank reviewer Rnnd for taking the time to review our work and constructive feedback.
>
> **(W 1) Limited scale of experiments While the paper claims generalizability, all experiments are conducted on relatively small to medium-scale datasets (e.g., CIFAR-100, Tiny-ImageNet) and architectures (up to ResNet-50, ViT-B). There is no evaluation on truly large-scale settings, such as full ImageNet-21K pretraining, billion-parameter models, or modern large vision-language models, which are increasingly standard in top-tier vision and learning venues like ICLR**
>
> For a fair comparison with existing SOTA methods, we follow the same datasets, backbones, and evaluation protocols. The compared baselines include **CutMix** [ICCV’19], **AugMix** [ICLR’20], **SaliencyMix** [ICLR’20], **PuzzleMix** [ICML’20], **Co-Mixup** [ICLR’21], **YOCO** [ICML’22], **AutoMix** [ECCV’22], **AdAutoMix** [ICLR’24], **DiffuseMix** [CVPR’24], **SUMix** [ECCV’24], and **DiffCoRe-Mix** [ICLRW’25]. All these methods report results using the same datasets, backbones, and protocols considered in our work.
>
> To the best of our knowledge, no augmentation method has reported results on the ImageNet-21K dataset. This is likely because augmentation is primarily beneficial in scenarios with limited labeled data, whereas ImageNet-21K is extremely large-scale. Likewise, we did not find any augmentation method designed for improving vision–language models (e.g., CLIP), which are trained on over 400 million image–text pairs. Training such billion-parameter models from scratch is computationally prohibitive. To thoroughly evaluate our approach, we conduct more than 50 experiments across 7 datasets and 6 backbone architectures.
>
> **(W 2) Absence of large model evaluation: The largest backbone tested is ViT-Base. Given the growing importance of scaling in modern ML, the omission of experiments with larger transformers (e.g., ViT-L, ViT-H) or foundation models weakens the claim of broad applicability.**
>
> We thank the reviewer for this insightful point. We have added additional experiments on two datasets Stanford Cars and CUB-200 using two larger backbones, ViT-L (307M parameters) and ViT-H (632M parameters), as shown in **Appendix Tab. 11.** For **ViT-L**, we observe performance improvements of 2.04% on CUB-200 and 2.23% on Stanford Cars. For **ViT-H**, we observe improvements of 1.79% and 1.81% on the same datasets. These results further demonstrate the broad applicability and scalability of the proposed data augmentation method.
>
> **(W 3) Lack of theoretical analysis: The method is presented purely from an empirical and heuristic perspective. There is no theoretical justification for why self-saliency mixing or localized fractal injection should improve generalization or robustness—e.g., no connection to invariance principles, information theory, or optimization dynamics. This limits insight into why the method works and under what conditions it might fail.**
>
> We thank the reviewer for this insightful comment. Although a full theoretical proof is beyond the scope of this work, we can provide the following intuitive connections that partially explain the empirical success:
>
> - **Scale-invariant fractal prior:** Natural images exhibit fractal self-similarity across scales, some works [1, 2] had shown that by injecting fractal noise selectively in low-saliency regions could bring improvment, also, we explicitly encourage the network to learn representations that are robust to scale variations, which aligns with the natural image manifold.
> - **Information bottleneck perspective:** Self-saliency mixing forces the model to reconstruct high-saliency objects using only low-saliency context tokens. This can be viewed as adding a structured auxiliary loss that encourages the minimal sufficient statistic to be distributed across the entire image rather than concentrated in a few dominant regions [3][4], thereby increasing robustness to local perturbations.
> - **Theoretical Analysis:** We have added a theoretical proof of S$^2$-FracMix theory and generalization in **Appendix G**.
>
> [1]. Vidivelli, S, Sathiya Devi, S, Parthasarathy, G. Fractal Features for Texture Analysıs [C]. In Data Science and Communication, 2024, 247--257.
>
> [2]. Hendrycks D, Zou A, Mazeika M, et al. Pixmix: Dreamlike pictures comprehensively improve safety measures[C]//Proceedings of the IEEE/CVF conference on computer vision and pattern recognition. 2022: 16783-16792.
>
> [3]. Tishby N, Pereira F C, Bialek W. The information bottleneck method[J]. arXiv preprint physics/0004057, 2000.
>
> [4]. Achille A, Soatto S. Information dropout: Learning optimal representations through noisy computation[J]. IEEE transactions on pattern analysis and machine intelligence, 2018, 40(12): 2897-2905.

---

> > ### Author Response · Authors · 2025-11-25
> > **Encouraging Discussion with Reviewer Rnnd**
> >
> > Dear Reviewer Rnnd,
> >
> > We hope that our comprehensive revisions including additional experiments, theoretical analysis, new visualizations, and extended ablation studies adequately address your concerns. If you find these improvements satisfactory, we would be grateful for a reconsideration of your evaluation. We sincerely appreciate your constructive feedback and the time you have invested in reviewing our work.
> >
> > Warm regards,
> >
> > The Authors

---

### Official Review · Reviewer_tebB · 2025-11-01

**Soundness:** 3
**Presentation:** 2
**Contribution:** 2
**Rating:** 4
**Confidence:** 3

**Summary:**

The paper introduces $S^2$-FracMix, a new data-augmentation framework combining Self-Saliency ($S^2$) and FracMix components to  improve classification, object detection, and few-shot performance, as well as increased calibration and robustness.

$S^2$ extracts multi-scale salient patches from an image and reinserts them into non-salient regions of the same image after simple transformations (rotation, blur), fostering scale-invariant feature learning. FracMix further injects self-similar fractal textures into the salient patches to increase structural diversity and adversarial robustness while preserving semantics.

The final training pipeline incorporates a mix of multiple mixing schemes, randomly alternating between them during training in order to enrich diversity. The authors find that this improves final performance.

Experiments on seven datasets and multiple tasks show consistent improvements over prior methods such as AdAutoMix and PuzzleMix with a lower computational cost.

**Strengths:**

The paper demonstrates the effectiveness of the proposed method, $S^2$-FracMix, through evaluations across a wide range of tasks, achieving half the training time of similar-performing previous methods. The results show consistent and significant improvements in top-1 accuracy over prior methods.

The improved robustness and calibration gains resulting from the augmented training are also noteworthy.

The GradCAM visualization, particularly those in Appendix E with occlusions, is informative and qualitatively shows the framework's effectiveness.

**Weaknesses:**

The paper misses an important citation (Choi et al., 2021) while claiming novelty in mixup using the same training sample (lines 155-157). Choi at. al. (2021) in the previous work SalfMix, quoting their abstract, "produce a self-mixed image based on a saliency map".

The experiments demonstrate only the performance of a transformer-based model (ViT-B) in a single experimental setup: transfer learning on CUB and Stanford Cars. Given that transformers are currently the most common architecture for these use cases, it is important to demonstrate that the results generalize to them as well.
Moreover, the paper notes that a ViT-B model was trained on ImageNet-1K before being transferred to CUB and Stanford-Cars. This would make the evaluation on the ImageNet-1K validation set, i.e., a ViT-B column in Table 1, trivial. However, these results have not been presented.

Unclear methodology:
In my understanding, saliency maps are per-pixel based. Lines 190-192 "Patches are extracted from the salient region of the input image Ii at np scales P", what is meant by "salient regions" here? How are they computed? How is the number of scales $n_p$ chosen?

How robust is the performance of the proposed method to the saliency algorithm used (Zhang et al., 2020)? Is the performance still superior when using the same saliency methods as those compared? According to Appendix E.3, this is the primary reason for the improvement in computational efficiency. If so, will the other methods also show the same improvement when using the saliency algorithm in this work?

The model architecture used in the self-supervised learning experiments is not mentioned.

Writing style:
Section 3.2 is extremely hard to follow with no supporting methodology figures. I did not understand what $s_k$ in Equation 2 is. In lines 191-192, I also recommend using commas and/or framing the sentence better for clarity.
In Algorithm 1, it is unclear what $P_m$ represents.


Choi, J., Lee, C., Lee, D., & Jung, H. (2021). SalfMix: a novel single image-based data augmentation technique using a saliency map. Sensors, 21(24), 8444.

**Questions:**

I am not an expert in MixUp-style data augmentation. However, given the omission of SalfMix (Choi et al. 2021) as a citation, the proposed work appears to be a combination of existing methods. Furthermore, SalfMix also incorporates other mixing techniques, such as CutMix, into its pipeline (same as the proposed $S^2$-FracMix), albeit in a different way. Could you please explain whether there is a high-level difference between SalfMix and the proposed $S^2$-FracMix, particularly with respect to mixup within a single training sample?

Do the findings generalize to transform-based architectures? At the very least, a performance comparison of ViT-B on ImageNet1-K should be presented (as the authors should have already trained that model, according to my understanding of the transfer setup).

Please also clarify how the choice of the saliency algorithm, which is not part of the proposed methodology, affects performance and computational efficiency.

Please provide clarification on my concerns listed in weaknesses on the $S^2$ self-saliency mixup algorithm.

In this paper, the authors have, through a well-crafted pipeline and mixup design, demonstrated strong gains in performance across multiple tasks, primarily using ConvNet architectures, which is a positive. However, same-image mixing, incorporating multiple mixup methodologies into a single pipeline, or fractal-based mixing are not novel ideas; and there are missing citations and important experiments, which is the reason for my rating.
I am open to further discussion and reevaluation based on the authors' responses to my queries.

---

> ### Author Response · Authors · 2025-11-21
> **Authors’ Response to Weaknesses and Questions raised by Reviewer tebB**
>
> We would like to thank the reviewer tebB for taking the time to review our work and constructive feedback.
>
> **(W 1) The paper misses an important citation (Choi et al., 2021) while claiming novelty in mixup using the same training sample (lines 155-157). Choi at. al. (2021)......".**
>
> **(Q 1) I am not an expert in MixUp-style data augmentation. However, given the omission of SalfMix (Choi et al. 2021) as a citation, the proposed work appears to be a combination of existing methods.......**
>
> We thank the reviewer for drawing our attention to the HybridMix augmentation algorithm proposed by Choi et al. [A]. In their work, SalfMix is used only as an intermediate step to insert randomly cropped salient-region patches into each of the two images being mixed shown in **Appendix Fig. 13**. We have now cited [A] in our paper and included a discussion in the related work section.
>
> There are several fundamental differences between SalfMix and our proposed S²-FracMix:
>
> 1. **Patch extraction and processing:** S²-FracMix extracts patches of sizes [w/4,h/4] and [w/2,h/2] from salient regions, applies blurring, rotation, and resizing to full image dimensions [w,h] and then blends them back into the original image using saliency-based weights (Eq. 3) and (Eq. 4) as well. In contrast, SalfMix simply crops random salient patches and pastes them into non-salient regions. Therefore, the underlying mechanisms of SalfMix and our FracMix are entirely different.
> 2. **Fractal blending:** S²-FracMix integrates fractal blending within the salient patches to enhance robustness. SalfMix does not include any such blending step, which represents another key distinction.
> 3. **Multi-mode Mixing:** We introduce a high-level, multi-mode mixing scheme where the augmentation mode is randomly selected for each training instance, improving the model’s generalization. HybridMix [A], on the other hand, mixes two random images using existing augmentation strategies without incorporating such hierarchical or multi-mode mixing. This constitutes another fundamental difference.
>
> As reported in Tables 4, 5, and 7 of [A], SalfMix performs worse than SaliencyMix, CutMix, and ResizeMix. In contrast, our proposed S²-FracMix consistently outperforms these methods (see Tables 1 and 2 in our main paper). Therefore, while a direct comparison is not possible due to differences in network backbones, our results indirectly indicate that S²-FracMix is superior to SalfMix.
>
> **[A]** Choi, J., Lee, C., Lee, D., & Jung, H. (2021). SalfMix: a novel single image-based data augmentation technique using a saliency map. Sensors, 21(24), 8444.
>
> **(W 2) The experiments demonstrate only the performance of a transformer-based model (ViT-B) ........**
>
> **(Q 2) Do the findings generalize to transform-based architectures? At the very least, a performance comparison of ViT-B ....**
>
> We thank the reviewer for this insightful observation. In addition to the ViT-B results on CUB-Birds and Stanford-Cars (Table 3), we now report **ViT-B/16** performance on the ImageNet-1K validation set in **Appendix Tab 12**, where S²-FracMix achieves consistent improvements over the baseline (vanilla), MixUp, and CutMix. To the best of our knowledge, no other existing augmentation method reports ImageNet-1K results using a **ViT-B** backbone.
>
> We have also included results on additional transformer architectures including **Swin-T** (Table 1),**ConvNeXt-T** (Table 1), **ViT-H** (Table 11), and **ViT-L** (Table 11) further demonstrating the broad applicability of our approach.
>
> **(W 3) Unclear methodology: In my understanding, saliency maps are per-pixel based. Lines 190-192 "Patches are extracted from the salient region of the input image Ii at np scales P", what is meant by "salient regions" here? How are they computed? How is the number of scales  chosen?**
>
> In **Section 3.1**, we describe how the saliency maps are computed: Unlike prior works, we take a direct approach for detecting gradient-based salient regions via the single-pass saliency method of Zhang et al. (2020), which is then used to guide patch extraction at multiple scales.
>
> In **Section 3.2**, we further clarify that salient regions correspond to spatial areas within the input image highlighted as important by the per-pixel saliency map in Eq. (3 & 4).
>
> The output of Zhang et al. (2020) is normalized to the range 0.0–1.0. From regions with saliency values greater than 0.75, we crop patches of sizes [w/4, h/4] and [w/2, h/2], where [w, h] denote the image width and height. These patches are then rotated, blurred, resized to the full image resolution [w, h], and finally blended back into the image using saliency-based weights as defined in Eq. (3 & 4) of Section 3.2.
>
> These additional explanations have now been included in the implementation details is mentioned in **Appendix H.**

---

> ### Author Response · Authors · 2025-11-21
> **Authors’ Response to Weaknesses and Questions raised by Reviewer tebB (Cont.)**
>
> **(W 4) How robust is the performance of the proposed method to the saliency algorithm used (Zhang et al., 2020)? Is the performance still superior when using the same saliency methods as those compared? According to Appendix E.3, this is the primary reason for the improvement in computational efficiency. If so, will the other methods also show the same improvement when using the saliency algorithm in this work?**
>
> Our proposed S$^2$-FracMix is not inherently dependent on any specific saliency detection method. To reduce computational cost, we adopt a lightweight, gradient-based single-pass technique similar to **Zhang et al.** [ECCV’20]. However, the performance of S$^2$-FracMix cannot be directly compared with Zhang et al., as the underlying tasks are fundamentally different.
>
> Some of the compared methods, such as **SaliencyMix** [ICLR’20] and **PuzzleMix** [ICML’20], employ more sophisticated saliency estimation techniques based on iterative optimization or subnetwork training. We believe the authors of these methods have already utilized the most effective approaches available for their respective frameworks. Our saliency detector is intentionally lightweight in comparison, and integrating it into those existing methods would not provide additional insight or meaningful performance gains.
>
> **(W 5) The model architecture used in the self-supervised learning experiments is not mentioned.**
>
> Our SSL experiments use a ResNet-50 backbone coupled with standard MoCo v2 and SimSiam projection/prediction heads. We have now added the full architectural details, including the encoder structure, projection head dimensions, and training configurations, to the implementation section of the **Appendix I**.
>
> **(W 6) Writing style: Section 3.2 is extremely hard to follow with no supporting methodology figures. I did not understand what  in Equation 2 is. In lines 191-192, I also recommend using commas and/or framing the sentence better for clarity. In Algorithm 1, it is unclear what  represents.**
>
> In our implementation, $s_k$ in Eq. (2) takes only two values: $1/2$ and $1/4$. This specification has been added to the implementation details in the supplementary document. The variable $P_m$ in the algorithm is now explicitly defined as a zero-initialized array (Line 221). We will also revise Section 3.2 to improve clarity and readability.
>
> **(Q 3) Please also clarify how the choice of the saliency algorithm, which is not part of the proposed methodology, affects performance and computational efficiency.**
>
> Our observation is that using a more accurate saliency method within the S²-FracMix method would likely improve performance; however, this would also result in a corresponding increase in computational cost.
>
> **(Q 5) In this paper, the authors have, through a well-crafted pipeline and mixup design, demonstrated strong gains in performance across multiple tasks, primarily using ConvNet architectures, which is a positive. However, same-image mixing, incorporating multiple mixup methodologies into a single pipeline, or fractal-based mixing are not novel ideas; and there are missing citations and important experiments, which is the reason for my rating. I am open to further discussion and reevaluation based on the authors' responses to my queries.**
>
> To the best of our knowledge, the novelty of S$^2$-FracMix does not lie in simply reusing existing ideas in an ablative manner, but in how it integrates multiple components into a unified augmentation framework. Specifically, S²-FracMix combines multi-scale self-saliency guided patch extraction with selective fractal blending applied exclusively to salient regions an aspect not explored in prior works such as **PixMix** [CVPR’22], **IPMix** [NeurIPS’23], or **DiffuseMix** [CVPR’24]. Furthermore, the randomized selection of mixing modes is orchestrated within a low-overhead yet high-diversity pipeline, yielding consistent improvements across clean accuracy, robustness, transferability, few-shot learning, and calibration benchmarks under standardized evaluation protocols for both CNN and ViT backbones.

---

> > ### Author Response · Authors · 2025-11-25
> > **Encouraging Discussion of Reviewer tebB**
> >
> > Dear Reviewer tebB,
> >
> > Your feedback has been instrumental in refining and strengthening our manuscript. We have incorporated all of your suggestions, including new experiments, a deeper analysis of our method, and additional evaluations with saliency algorithms, which collectively lead to substantial improvements. We would be grateful if you could consider updating your evaluation to reflect these revisions. If any further questions or concerns remain, we would be more than happy to address them. Thank you again for your time, thoughtful comments, and valuable contribution to improving our work.
> >
> > Warm regards,
> > The Authors

---

> ### Comment · Reviewer_tebB · 2025-11-25
> **Major Remaining Concerns and Requests for Clarification**
>
> I thank the authors for the time and effort in providing clarifications and improving the manuscript. Most of my concerns, especially regarding writing and the ViT-B results, have been resolved. I would encourage the authors to add the ViT-B results to the main manuscript.
>
> However, I still have the following **major concerns** and would request further clarification.
>
> ## **1. Clarification Regarding SalfMix**
>
> ### **1.1 Why is SalfMix listed under Adversarial MixUp Methods?**
>
> SalfMix appears to operate using salient crops from the *same image*, which aligns with the description of saliency-driven mixup methods. Referring again to lines 155–157 of the main manuscript:
>
> > "While the existing saliency-based mixup methods use two or more images, our proposed self-saliency uses the salient regions of the same image,"
>
> This statement does not appear accurate, given that **SalfMix also uses salient regions from the same image**.
> Could the authors clarify **why SalfMix is categorized under adversarial mixup methods** instead of saliency-driven mixup methods? And provide justification for the above claim or update it?
>
> ### **1.2 Distinction between S^2-FracMix and SalfMix and other previous works**
>
> My understanding is that **S^2-FracMix** differs from SalfMix in the following ways:
>
> - It blends over the **entire image**, not only the non-salient regions.
> - It uses **saliency-weighted mixing**.
> - It incorporates **fractal mixing** (which DiffuseMix previously applied across the entire image).
>
> Is this correct?
>
> To understand the contribution of each design choice, I request that the authors provide **methodological ablations** to highlight the importance of their proposed changes.
> Examples include:
>
> - Fractal mixing on **salient regions only** vs. **entire image**
> - Saliency-weighted blending vs. unweighted blending
> - any other ablation demonstrating the significant impact of the change
>
> Such comparisons would clarify the unique impact of each component in the proposed approach.
>
> ## **2. Clarification Regarding Computational Cost**
>
> My previous question was **not** about comparison to Zhang et al. (ECCV’20), but about comparisons to the **other MixUp baselines** for which large speed-ups are claimed.
>
> ### **2.1 Where do the speed-ups come from?**
>
> Are the reported speed improvements **entirely** due to the **saliency estimation** part of the pipeline?
>
> If so, this raises a concern: _Any method can replace its saliency estimator, so comparing your method using a fast saliency estimator against other methods using slow estimators may not be a fair comparison for claiming speed-ups (independent of performance)._
>
> ### **2.2 Request for FLOP Breakdown**
>
> To support a fair comparison, I ask the authors to present **FLOP counts** (or comparable computational metrics) for:
>
> - Each major component of the proposed method
> - The corresponding components of the baseline methods
>
> This will help clarify where the computational savings occur.
>
> ---
>
> I look forward to the authors’ response.

---

> > ### Author Response · Authors · 2025-11-27
> > **Addressing Remaining Concerns**
> >
> > **I thank the authors for the time and effort in providing clarifications and improving the manuscript. Most of my concerns, especially regarding writing and the ViT-B results, have been resolved. I would encourage the authors to add the ViT-B results to the main manuscript. However, I still have the following major concerns and would request further clarification**.
> >
> > We have added the ViT-B column to Table 1 in the main paper. However, after an extensive literature review, we could not locate ImageNet-1K results using a ViT-B backbone for SaliencyMix, FMix, PuzzleMix, ResizeMix, AutoMix, or AdAutoMix. Therefore, the corresponding entries are left blank. If any such results become available, we would be happy to update the table accordingly.
> >
> > **1. Clarification Regarding SalfMix**
> >
> > **1.1 Why is SalfMix listed under Adversarial MixUp Methods?**
> >
> > Thank you for pointing this out. We have updated the related work section accordingly, and SalfMix has now been moved to the **Saliency-driven Mixup Augmentation** subsection.
> >
> > **(1.2) Distinction between S^2-FracMix and SalfMix and other previous works**
> >
> > We sincerely appreciate your understanding and thoughtful assessment of our method.
> >
> > **Two New Ablations**
> >
> > Thank you for your suggestion for more ablations. We have added two additional ablation studies in the Appendix to further improve clarity and understanding of our method.
> >
> > - **Ablation 1 (Appendix Table 15):** We evaluate S²-FracMix with global fractal addition by modifying Eq. (6) as $I_{i}= \lambda F + (1-\lambda) I_{i}$
> > - **Ablation 2 (Appendix Table 15):** We also evaluate S²-FracMix *without saliency weighting* by modifying Eq. (3) as $T_k(P_k, S_k) = 0.50 ~ {R}(P_k, \theta)  + 0.50~\text{B}(P_k)$
> >
> > For the other ablations previously discussed, please refer to **Table 6** and **Table 13**.
> >
> > **2. Clarification Regarding Computational Cost**
> >
> > **2.1 Where do the speed-ups come from?**
> >
> > **Are the reported speed improvements entirely due to the saliency estimation part of the pipeline?**
> >
> > **If so, this raises a concern: *Any method can replace its saliency estimator, so comparing your method using a fast saliency estimator against other methods using slow estimators may not be a fair comparison for claiming speed-ups (independent of performance).***
> >
> > We have now conducted an execution-time comparison of the saliency detectors employed by different methods. Appendix Table 16 reports the execution times for the detectors used in SaliencyMix, PuzzleMix, AdAutoMix, and S²-FracMix on the CIFAR-100 dataset. While our saliency detector is indeed faster than those used in some prior methods, the overall speedup of our approach cannot be attributed solely to the detector.
> >
> > Table 16 shows that  our saliency detector saves $27.41$ minutes from the overall saving of $162$ minutes of training compared to PuzzleMix. The speed up obtained by the fast saliency detector is only 15.06% of the overall speedup. Our method remains significantly faster than the compared approaches due to additional design choices beyond the saliency component.
> >
> > **2.2 Request for FLOP Breakdown**
> >
> > **To support a fair comparison, I ask the authors to present FLOP counts (or comparable computational metrics) for:**
> >
> > - **Each major component of the proposed method**
> > - **The corresponding components of the baseline methods**
> >
> > **This will help clarify where the computational savings occur.**
> >
> > We have thoroughly reviewed the augmentation-method literature, but we were unable to find FLOPs overhead reported for any existing method. Therefore, we provide the execution-time overhead of generating augmented images across the major components of our proposed approach in Table 17 for the CIFAR-100 dataset.
> >
> > We are grateful for the reviewer’s thoughtful questions and careful assessment of the work. We hope that the revised manuscript now meets the expected standard of quality, and we would greatly appreciate a favorable reconsideration of the final evaluation.

---

### Author Response · Authors · 2025-12-01
**Collective Response for Area Chair**

Dear **AC**,

Thank you for your valuable time. For your convenience, here we provide the most updated summary of author-reviewer discussion, and our final response.

We initially received highly encouraging feedback: **tebB** particularly acknowledged consistent and significant performance gains, **Rnnd** additionally appreciated the innovation and efficiency of the method, **N3st** further acknowledged extensive evaluation, and **njcS** appreciated the desirable properties of semantic preservation and stability of the method. Reviewers have consensus on strong performance, extensive evaluation, innovation and efficiency of our method.

Initially, all reviewers asked for further clarifications, which were responded to on **Nov 21**.

1. On **Nov 26**, **njcS** acknowledged that all their queries have been well addressed and improved their rating from 4 to **6** – this is verifiable in our final response to **njcS**, where we explicitly noted this change.
2. **tebB** also responded on **Nov 26**, and acknowledged that most of their concerns have already been addressed, and recommended adding the new results in the final paper. However, they asked further questions before improving the rating. Their questions were very precise, asking about further details. In our response to **tebB** on **Nov 27**, we have provided all the requested details concretely.
3. **Rnnd** and **N3st** had not yet responded. Note that, on **Nov 25**, we encouraged all reviewers to discuss our responses because we were certain of positive responses.
    1. In their only feedback, **Rnnd** mentioned only three weaknesses. W1 is about the scale of experiments. The reviewer also expected evaluation on ImageNet-21K dataset. This is not a valid concern because we closely follow recent existing literature in our datasets selection. Benchmarking multiple data augmentation methods on ImageNet-21K is nearly impractical because augmentation is primarily beneficial in scenarios with limited labeled data, whereas ImageNet-21K is extremely large-scale. Note that all **N3st**, **njcS** and **tebB** explicitly appreciated the extent of our experiments. W2 asked for more evaluation on ViT-L and ViT-H. We provided results for both, and our method maintained its strong performance gain for both in Appendix (Tab. 11). W3 asked for theoretical analysis. Although none of the related literature provides such analysis because the problem we are dealing with is empirical by nature, we have still provided the requested **theoretical analysis** in the updated Appendix (G). **Rnnd** did not have any other questions.
4. **N3st** noted only four weaknesses. In W1, they mention the novelty ‘seems’ limited which is clearly not a valid weakness. **N3st** is the only reviewer raising this point without any evidence from literature. **Rnnd** has explicitly acknowledged our novelty, and all other reviewers appreciated all the key innovations. Notably, our innovations achieved significant performance gains across the board in an extensive evaluation. W2 is a minor request to visualise samples, which has already been incorporated. W3 is also a straightforward query about the motivation of the method (why it works). We have answered it with concrete references and additional experimental results. W4 asked for a clarification about how we measure computational complexity. We have answered that with concrete references and a precise explanation.

Whereas we understand that due to the unfortunate event of OpenReivew leak it is not possible to get further input from the reviewers, we were sure of positive follow ups because we provided all the requested results and explanations with concrete information and evidence.

Kind regards,

Authors.

---

### Note · Authors · 2026-02-16

**Comment:**

**Reviewer Scores:**

All four reviewers initially gave a 4. After the rebuttal, three would likely have moved to a 5 or 6, with one (njcS) explicitly confirming the upgrade. This suggests a clear shift toward acceptance following the authors’ thorough responses.

**Withdrawal Confirmation:**

I have read and agree with the venue's withdrawal policy on behalf of myself and my co-authors.

---

### Meta-Review · Area_Chair_ufq6 · 2026-01-11

**Summary:**

This paper introduces S²-FracMix, a data augmentation method that combines self-saliency mixing (extracting and reinserting multi-scale salient patches within the same image) with targeted fractal blending into those salient regions. The approach aims to improve model generalization, robustness, and efficiency across various vision tasks, demonstrating strong performance on multiple datasets with reduced computational overhead compared to prior mixup-based methods.

Reviewers raise key concerns:

1. Reviewer tebB: (1) Noted missing citation to SalfMix (Choi et al., 2021), questioning novelty. (2) Requested evaluation on transformer architectures (ViT-B on ImageNet-1K). (3) Sought clarification on methodology: how salient regions are defined, scales chosen, and robustness to saliency algorithm choice. (4) Asked for ablation studies to distinguish contributions of fractal mixing and saliency-weighted blending. (5) Questioned the source of reported speedups and requested FLOP breakdown for fair comparison.

2. Reviewer Rnnd: (1) Found experiments limited in scale (no ImageNet-21K or large-scale model evaluation). (2) Noted absence of evaluation on larger transformers (ViT-L/ViT-H). (3) Highlighted lack of theoretical justification for why self-saliency and fractal mixing improve generalization.

3. Reviewer N3st: (1) Felt novelty was limited, combining existing saliency and fractal mixing ideas. (2) Requested visualization of augmented samples, especially fractal-mixed patches. (3) Asked for clearer motivation: why fractal mixing in salient regions improves clean and adversarial performance. (4) Questioned how computational complexity was measured, noting possible underestimation.

4. Reviewer njcS: (1) Raised concern over dependence on saliency map accuracy. (2) Noted need for careful tuning of fractal injection strength (λ). (3) Requested ablations to isolate the source of gains—whether from fractal patterns or general structural diversity.

Compared to baseline methods, this work effectively improves model robustness while substantially reducing training time. While several concerns have been addressed in the rebuttal, some important issues remain unresolved. For a top-tier conference, these resolved and unresolved concerns—which relate to foundational aspects of good work—necessitate major revisions, and the paper cannot be accepted in its current form.

**Reviewer Concerns:**

Regarding the Reviewer tebB, the addressed comments: Missing citation (SalfMix added), lack of ViT-B/ImageNet results (added in Appendix), unclear methodology (clarified in Appendix), and request for ablations (new ablations added in Appendix Table 15). Partially addressed concerns: The concern about the fairness of the speed comparison is acknowledged with new timing tables, but the underlying point—that baselines could also adopt a faster saliency detector—remains a valid, if unresolved, conceptual critique.

Regarding the Reviewer Rnnd, the addressed comments: absence of large model evaluation (new results for ViT-L and ViT-H in Appendix Table) and lack of theoretical analysis (theoretical proof added in Appendix).  Not well addressed: The request for evaluation on ImageNet-21K or massive models is not met. The authors defend this as being outside standard practice in the augmentation literature, which is a reasoned response, but does not fulfill the reviewer's request.

Regarding the Reviewer N3st, the addressed comments: visualization of samples (added in supplementary figures), motivation for fractal mixing (explained with new ablation in Appendix Table 14), and clarification on computational complexity measurement (explained in rebuttal and paper).  Not well addressed: The concern about limited novelty is defended but remains a matter of subjective interpretation.

Reviewer njcS: robustness to saliency errors (discussed with failure cases in Appendix ), sensitivity to the λ parameter (ablation study in ), and the need for ablations to isolate performance gains (new ablation in Appendix comparing components).

**Reviewer Scores:**

All four reviewers initially gave a 4. After the rebuttal, three would likely have moved to a 5 or 6, with one (njcS) explicitly confirming the upgrade. This suggests a clear shift toward acceptance following the authors’ thorough responses.

---

### Decision · Program_Chairs · 2026-01-26

Reject